# Cannabidiol sensitizes TRPV2 channels to activation by 2-APB

**Aaron Gochman**[1†‡], **Xiao-Feng Tan**[1†], **Chanhyung Bae**[1§], **Helen Chen**[2], **Kenton J Swartz**[1], **Andres Jara-Oseguera**[2*†]

[1]Molecular Physiology and Biophysics Section, Porter Neuroscience Research Center, National Institute of Neurological Disorders and Stroke, National Institutes of Health, Bethesda, United States; [2]Department of Molecular Biosciences, College of Natural Sciences, The University of Texas at Austin, Austin TX, United States

**\*For correspondence:**
andres.jaraoseguera@austin.utexas.edu

[†]These authors contributed equally to this work

**Present address:** [‡]Vanderbilt University School of Medicine, Nashville, United States; [§]Janssen R&D, Biologics Discovery, Spring House, United States

**Abstract** The cation-permeable TRPV2 channel is important for cardiac and immune cell function. Cannabidiol (CBD), a non-psychoactive cannabinoid of clinical relevance, is one of the few molecules known to activate TRPV2. Using the patch-clamp technique, we discover that CBD can sensitize current responses of the rat TRPV2 channel to the synthetic agonist 2-aminoethoxydiphenyl borate (2-APB) by over two orders of magnitude, without sensitizing channels to activation by moderate (40°C) heat. Using cryo-EM, we uncover a new small-molecule binding site in the pore domain of rTRPV2 in addition to a nearby CBD site that had already been reported. The TRPV1 and TRPV3 channels are also activated by 2-APB and CBD and share multiple conserved features with TRPV2, but we find that strong sensitization by CBD is only observed in TRPV3, while sensitization for TRPV1 is much weaker. Mutations at non-conserved positions between rTRPV2 and rTRPV1 in either the pore domain or the CBD sites failed to confer strong sensitization by CBD in mutant rTRPV1 channels. Together, our results indicate that CBD-dependent sensitization of rTRPV2 channels engages multiple channel regions, and that the difference in sensitization strength between rTRPV2 and rTRPV1 channels does not originate from amino acid sequence differences at the CBD binding site or the pore domain. The remarkably robust effect of CBD on TRPV2 and TRPV3 channels offers a promising new tool to both understand and overcome one of the major roadblocks in the study of these channels – their resilience to activation.

## Editor's evaluation

This is an important report on the discovery of a strong sensitizing effect of cannabidiol on the activation of TRPV2 channels by 2-APB. The conclusions are convincingly supported by electrophysiological recordings and cryo-EM structures, but identification of a clear molecular mechanism will require additional structural work. The paper will be of interest to the ion channel research community.

## Introduction

The non-selective, cation-permeable transient receptor potential vanilloid 2 (TRPV2) channel is a homo-tetrameric protein (*Huynh et al., 2016*; *Zubcevic et al., 2016*) expressed in multiple cell types and tissues in animals (*Caterina et al., 1999*; *Tamura et al., 2005*). Adult mice in which TRPV2 channel expression in cardiomyocytes was ablated died within 2 wk and exhibited severe structural abnormalities of the heart (*Iwata and Matsumura, 2019*). In neonatal mice, TRPV2-deficient cardiomyocytes were also highly dysfunctional after growth in culture (*Katanosaka et al., 2014*), indicating that TRPV2 channel expression in cardiomyocytes is required for normal development and function of the heart.

In macrophages, TRPV2 channel expression is essential for phagocytosis (*Link et al., 2010*; *Lévêque et al., 2018*), in pancreatic β-cells it influences insulin secretion (*Hisanaga et al., 2009*), in red blood cells it contributes to their response to osmotic challenges (*Belkacemi et al., 2021*), and its expression levels are significantly altered in multiple types of cancers (*Monet et al., 2010*; *Yamada et al., 2010*; *Kudou et al., 2019*; *Siveen et al., 2020*; *Guéguinou et al., 2021*).

Interestingly, the identity of the stimuli that orchestrate TRPV2 channel activity in each of the examples mentioned above remains unknown. Very few stimuli capable of activating the TRPV2 channel have been identified; the extreme temperatures (>55°C) required to activate rodent TRPV2 channels (*Caterina et al., 1999*; *Tamura et al., 2005*; *Yao et al., 2011*; *Liu and Qin, 2016*) and the lack of thermo-sensitivity of the human orthologue under basal conditions (*Neeper et al., 2007*; *Yao et al., 2011*) exclude a likely role for this protein in thermo-detection. Yet, oxidative modification of specific methionine residues in rodent and human TRPV2 channels resulted in robust channel activation at temperatures <50°C (*Fricke et al., 2019*), suggesting that redox processes, temperature, and membrane depolarization together could promote channel activity in native tissues under physiological or pathological conditions. The synthetic agonist 2-aminoethoxydiphenyl borate (2-APB) robustly activates rodent but not human TRPV2 channels (*Hu et al., 2004*; *Juvin et al., 2007*), and even rodent TRPV2 channels have very low apparent affinity for 2-APB, with an $EC_{50} > 1$ mM (*Gao et al., 2016a*; *Liu and Qin, 2016*) that is close to the solubility limit of the compound. Importantly, 2-APB also targets other channels, including TRP channels (*Hu et al., 2004*; *Xu et al., 2005*; *Togashi et al., 2008*; *Kovacs et al., 2012*), STIM-Orai channels (*Bootman et al., 2002*), and gap-junctions (*Bai et al., 2006*). Cannabidiol (CBD), a non-psychotropic compound from the cannabis plant that has received much attention recently for its potential use in treating a variety of disorders (*Pauli et al., 2020*), has also been found to activate rodent TRPV2 channels with much higher apparent affinity than 2-APB (*Qin et al., 2008*). Other cannabis-derived compounds, including $\Delta^9$-tetrahydrocannabinol (THC), are also reported to activate TRPV2 channels (*Neeper et al., 2007*; *Qin et al., 2008*; *Zhang et al., 2022*).

In stark contrast with the limited number of stimuli that activate the TRPV2 channel, the TRPV1 channel expressed in nociceptive neurons can be activated by the stimuli that activate the TRPV2 channel, but also by a remarkably diverse set of inflammatory mediators (*Zygmunt al., 1999*; *Hwang et al., 2000*; *Shin et al., 2002*; *Nieto-Posadas et al., 2011*; *Joseph et al., 2019*), animal toxins (*Bohlen et al., 2010*; *Yang et al., 2015*), and natural products (*Caterina et al., 1997*; *Salazar et al., 2008*). Paradoxically, both channels are >40% identical in amino acid sequence, have an identical structural fold (*Gao et al., 2016b*; *Huynh et al., 2016*; *Zubcevic et al., 2016*), and utilize similar mechanisms to gate the passage of cations through their ion conduction pathway in response to stimulation with 2-APB (*Jara-Oseguera et al., 2019*). Notably, high sensitivity to a TRPV1-specific agonist, resiniferatoxin (RTx), could be engineered into the rat TRPV2 channel simply by substituting four non-conserved residues in a pocket that is otherwise equivalent to where RTx binds in TRPV1 (*Yang et al., 2016*; *Zhang et al., 2016*; *Zhang et al., 2019*).

Cryo-electron microscopy (cryo-EM) structures of rat TRPV2 channels obtained in the presence of CBD (*Pumroy et al., 2019*) and CBD together with 2-APB (*Pumroy et al., 2022*) suggest that the cannabinoid binds close to the regions in the channel that directly gate the passage of cations through the pore in response to stimulation. Here we set out to use the patch-clamp technique and cryo-EM structural determination to characterize the actions of CBD on rat TRPV2 channels, and learn more about how the mechanisms of TRPV2 channel activation contrast with those of its closest homologue, the TRPV1 channel that can be activated by so many different types of stimuli. We discovered that CBD potently sensitizes the rTRPV2 channel to activation by 2-APB, without an effect on channel sensitivity to moderate heat. We obtained structures of rTRPV2 in the presence of CBD and 2-APB representing two non-conducting conformations; one has CBD bound to a previously identified site (*Pumroy et al., 2019*; *Pumroy et al., 2022*) providing confirmatory evidence for the site. The other conformation exhibits an additional non-protein density at a site located at an allosteric nexus between the pore domain, the S4-S5 linker, and a bound lipid molecule. We assign this density to CBD, but it is possible that other molecules, including endogenous ligands, could occupy that site to modulate channel function physiologically. When comparing the effect of CBD on the TRPV1 and TRPV3 channels, which are also activated by CBD and 2-APB (*Hu et al., 2004*), we find that only TRPV3 channels are potently sensitized by CBD. Further, mutations at rTRPV1 sites that differ between TRPV1 and TRPV2 and that are located in the pore domain or the CBD binding region failed to confer

strong sensitization by CBD. These findings establish that the robust sensitization by CBD observed in TRPV2 channels involves a CBD-specific allosteric mechanism that engages channel regions distant from the CBD binding site and the pore.

## Results

### CBD strongly sensitizes rTRPV2 channels to activation by 2-APB

We expressed rat TRPV2 (rTRPV2) channels in HEK293 cells and began by measuring the magnitude of the currents elicited by a low concentration of 2-APB (0.5 mM) or a near-saturating concentration (10 µM) of CBD (*Qin et al., 2008*) in the whole-cell configuration of the patch clamp at a holding potential of –80 mV. Even at this low concentration, 2-APB elicited currents that were much larger than those by CBD (*Figure 1A*). When 2-APB and CBD were applied together, we observed large currents that were comparable to those measured in response to a concentration of 2-APB (6 mM) that maximally activates rTRPV2 channels (*Figure 1A*). Currents in the presence of 0.5 mM 2-APB were over two orders of magnitude larger in the presence of CBD than in its absence (*Figure 1H*). Importantly, 2-APB and CBD applied together elicited no increase in whole-cell currents from un-transfected cells recorded in response to voltage pulses from –100 to +100 mV (*Figure 1B*). In contrast, the same voltage-stimulation protocol elicited robust currents in rTRPV2-transfected cells when exposed to 0.5 mM 2-APB or 10 µM CBD applied separately (*Figure 1C and D*). The sensitization caused by CBD is so strong that it becomes challenging to quantitate; at 0.5 mM 2-APB, channel activity is barely detectable in the whole-cell configuration, and yet with CBD added currents reach maximal activation levels (*Figure 1A and H*). The magnitude of sensitization we measured (*Figure 1H*) likely represents a lower bound for the sensitizing effect of CBD on rTRPV2 channels. We therefore analyzed data in a semi-quantitative manner without attempting to quantify the energetics associated with sensitization.

We were surprised at the minimal efficacy with which CBD activates rTRPV2 channels (*Figure 1A, C and D*), so we measured the magnitude of currents elicited by increasing concentrations of CBD in rTRPV2-expressing cells. The resulting concentration–response relations (*Figure 1E*) confirm that CBD activates rTRPV2 channels with much higher affinity ($EC_{50} \sim 4$ µM) than 2-APB but also with much lower efficacy, and that rTRPV2 channels can be assumed to be fully bound by CBD at a concentration of 10 µM CBD. We next tested whether rTRPV2 channels display an increased apparent affinity for 2-APB when bound to CBD by measuring rTRPV2 channel activation at increasing concentrations of 2-APB in the presence and absence of 10 µM CBD, and found that the $EC_{50}$ for 2-APB activation decreased approximately tenfold in the presence of CBD (*Figure 1F*).

The TRPV2 channel is reported to undergo sensitization upon stimulation with 2-APB that is irreversible over the duration of a patch-clamp experiment, leading to hysteresis in the 2-APB dose–response relations obtained pre- and post-sensitization: in most but not all patches, repeated short exposures to low concentrations of 2-APB were found to elicit progressively larger currents until reaching a plateau ~15-fold larger than the initial response (*Liu and Qin, 2016*). Sensitization also reached saturation after a single short stimulation with a concentration of 2-APB that maximally activates channels (*Liu and Qin, 2016*). To investigate whether 2-APB and CBD sensitize rTRPV2 channels to a similar extent, we performed experiments where we first briefly exposed cells to 0.5 mM 2-APB, then we stimulated cells with a maximally activating concentration of 2-APB (6 mM), which we repeated three times to ensure all channels had become sensitized ($I_1$, $I_2$, and $I_3$, *Figure 1G*), and finally exposed cells to 0.5 mM 2-APB and 10 µM CBD first applied separately and then together. We found that currents activated by 0.5 mM 2-APB increased nearly tenfold after repeated stimulation with 6 mM 2-APB in some but not all cells (*Figure 1G*, ① vs. ②), whereas co-application with CBD increased currents by >100-fold in all cells (*Figure 1G and H*). The observed magnitude and cell-to-cell variability of 2-APB-dependent sensitization is consistent with previously reported measurements (*Liu and Qin, 2016*), and markedly smaller than sensitization by CBD. These results indicate that the predominant states adopted by rTPRV2 channels when bound to CBD must be energetically different than those adopted by 2-APB-sensitized channels in the absence of ligands.

Recording solutions containing 2-APB and CBD also included dimethyl sulfoxide (DMSO) that we used to make stock solutions of both compounds, so the amount of DMSO applied to cells was larger in solutions that contained both agonists. To rule out an influence of DMSO on our results, in the same experiments described above we exposed cells to solutions with either 2-APB or CBD with added

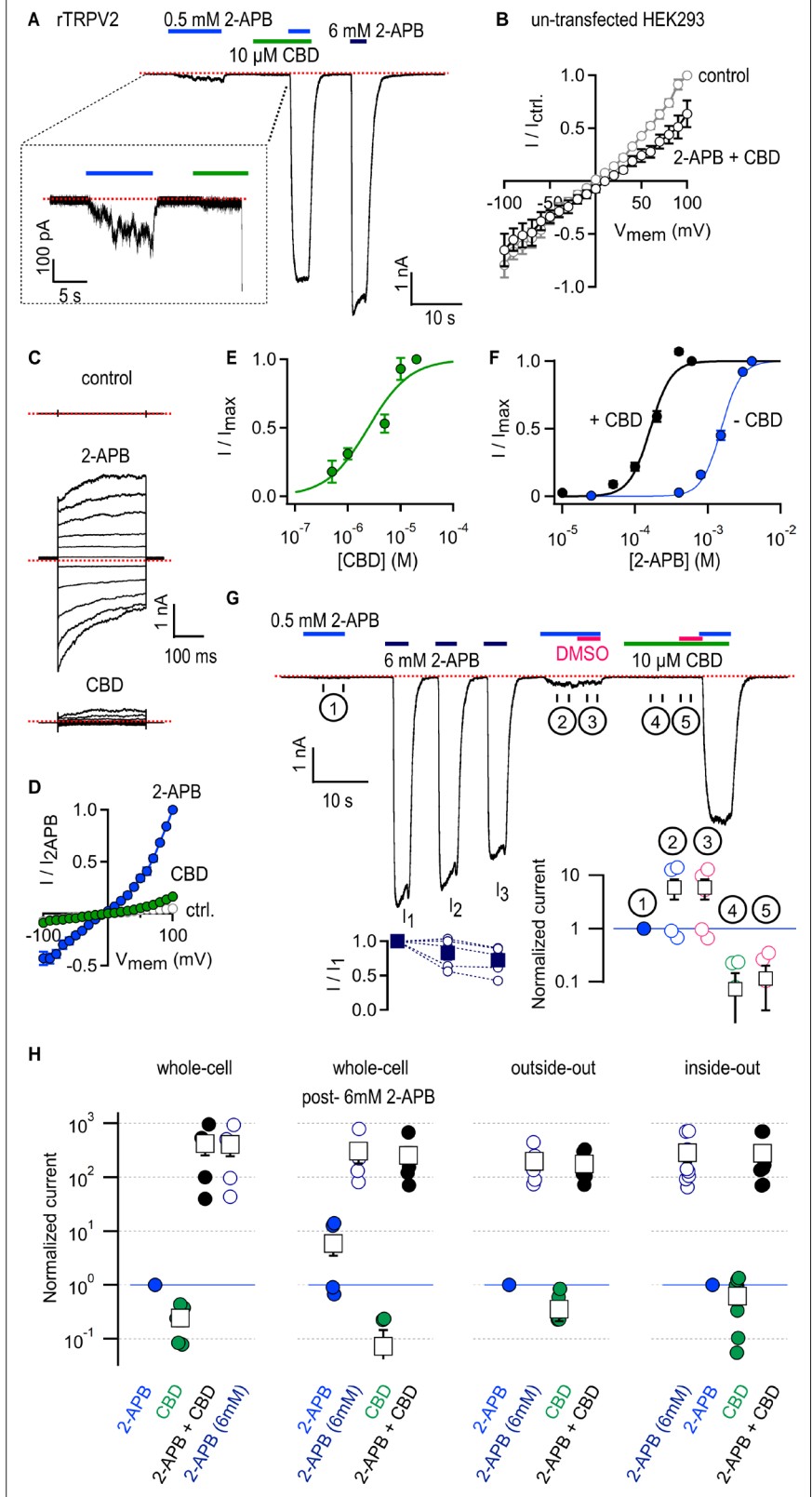

**Figure 1.** Cannabidiol (CBD) strongly sensitizes rTRPV2 channels to activation by 2-aminoethoxydiphenyl borate (2-APB) in whole-cell and in excised patch recordings. (**A**) Representative whole-cell gap-free current recording at –80 mV from a cell expressing rTRPV2 channels. The colored horizontal lines denote the duration of exposure to test compounds, and the red dotted line denotes the zero-current level. The inset shows a magnified view

*Figure 1 continued*

of a segment of the recording. (**B**) Mean current–voltage relations recorded in control solution and in the presence of 0.5 mM 2-APB + 10 µM obtained from un-transfected cells in the whole-cell configuration (n = 8). (**C**) Representative current families elicited by voltage steps from –100 to +100 mV obtained from an rTRPV2-expressing cell exposed to control solution, 0.5 mM 2-APB or 10 µM CBD. The dotted lines indicate the zero-current level. (**D**) Mean current–voltage relations obtained from data as in (**C**) and normalized to the mean value at +100 mV in the presence of 0.5 mM 2-APB. Data are shown as mean ± SEM (n = 5). (**E**) Dose–response relation for rTRPV2 channel activation by CBD measured at –80 mV in the whole-cell configuration (mean ± SEM; n = 4). The continuous curve is a fit to the Hill equation with parameters: $EC_{50}$ = 4.3 ± 1.4 µM and Hill coefficient ($n_H$) = 1.7 ± 0.1. (**F**) Concentration–response relations for rTRPV2 channel activation by 2-APB at –80 mV in the whole-cell configuration measured in the absence (blue symbols) or presence (black symbols) of 10 µM CBD (mean ± SEM; n = 7). Hill equation parameters: no CBD, $EC_{50}$ = 1.5 ± 0.05 mM, Hill coefficient ($n_H$) = 3.3 ± 0.2; 10 µM CBD, $EC_{50}$ = 159.9 ± 7.7 µM, Hill coefficient ($n_H$) = 2.9 ± 0.3. (**G**) Representative whole-cell gap-free recording at –80 mV in an rTRPV2-expressing cell. The bottom-left inset shows group data for the mean current amplitude at each of the three stimulations with 6 mM 2-APB, leak-subtracted and normalized to the amplitude during the first 6 mM 2-APB stimulation (solid squares – mean ± SEM, n = 5; empty circles – data from individual cells). The bottom-right inset displays steady-state data measured over each of the five intervals (① - ⑤) identified by the circled numbers on the current time course, normalized to the response to 2-APB in ① denoted by the horizontal blue line. Black squares are mean ± SEM, and circles are data from individual cells. (**H**) Leak-subtracted and normalized group data for: experiments in the whole-cell configuration as in (**A**) (n = 5) or post-sensitization by 6 mM 2-APB as in (**G**) (n = 5); experiments from outside-out patches as in *Figure 1—figure supplement 3A* (n = 6); experiments from inside-out patches as in *Figure 1—figure supplement 3B* (n = 9). Data was normalized to the first stimulation with 0.5 mM 2-APB (denoted by the horizontal blue lines) and shown as mean ± SEM (black squares) or values from individual cells (circles).

The online version of this article includes the following source data and figure supplement(s) for figure 1:

**Source data 1.** Excel file with group data from electrophysiological recordings shown in *Figure 1*.

**Figure supplement 1.** No evidence of chemical reactivity between 2-aminoethoxydiphenyl borate (2-APB) and cannabidiol (CBD).

**Figure supplement 2.** Cannabidiol (CBD) does not strongly sensitize rTRPV2 channels to activation by heat at physiological temperatures.

**Figure supplement 2—source data 1.** Current–temperature relations and $Q_{10}$ values.

**Figure supplement 3.** Sensitization of rTRPV2 channels by cannabidiol (CBD) observed in excised membrane patches.

DMSO so that its concentration was the same as in solutions that contained both agonists together. We found that the additional DMSO did not change the magnitude of the currents in 0.5 mM 2-APB (*Figure 1G*, ② vs. ③) or 10 µM CBD (*Figure 1G*, ④ vs. ⑤).

To rule out a direct chemical reaction between 2-APB and CBD, yielding a product with enhanced potency to activate the channel, we ran samples containing our recording solution (blank), 2-APB, CBD, and 2-APB and CBD mixed together, through a high-performance liquid chromatography (HPLC) system. We found that the elution time and height of the two peaks present in a sample with both agonists perfectly matched each single peak observed in samples with CBD or 2-APB alone, suggesting the two do not interact chemically (*Figure 1—figure supplement 1*).

In addition to being weakly sensitized by 2-APB, rTRPV2 channels are strongly and irreversibly sensitized by heat, in which channel activation with extreme heat resulted in a tenfold decrease in the $EC_{50}$ for 2-APB (*Liu and Qin, 2016*). In addition, heat-sensitized rTRPV2 channels no longer require extreme temperatures (>55°C) to activate, showing current responses at temperatures at or below 40°C (*Liu and Qin, 2016*). We therefore tested whether the mechanisms of rTRPV2 channel sensitization by CBD and heat are related. We reasoned that if this was the case, then the presence of CBD should facilitate channel activation by heat and enable current responses at around 40°C. Using rTRPV1 channel currents as a positive control for their high sensitivity to temperature changes in the 20–40°C range, we successfully detected steeply temperature-dependent currents as previously described (*Figure 1—figure supplement 2A and E*; *Caterina et al., 1997*; *Yao et al., 2010*; *Jara-Oseguera et al., 2016*). When we performed experiments using cells expressing rTRPV2 channels in the absence of CBD, we were unable to detect any temperature-dependent changes in current that were noticeably different from those measured under identical conditions in un-transfected cells

(*Figure 1—figure supplement 2B, C, E*) as expected because the threshold for heat activation for rTRPV2 channels is >55°C (*Yao et al., 2011*; *Liu and Qin, 2016*). Interestingly, the responses in rTRPV2-expressing cells were the same in the absence and presence of CBD (*Figure 1—figure supplement 2D and E*), indicating that CBD does not strongly sensitize rTRPV2 channels to activation by heat. We also tested whether CBD sensitizes rTRPV2 channels to activation by the synthetic agonist probenecid (*Bang et al., 2007*), but we failed to detect any probenecid-elicited currents in the absence or presence of CBD (data not shown).

Finally, we tested whether the sensitizing effect of CBD on activation of rTRPV2 channels by 2-APB also occurs in excised membrane patches devoid of many cellular components. Exposure of outside-out or inside-out patches expressing rTRPV2 channels to 0.5 mM 2-APB or 10 µM CBD alone elicited negligible currents (*Figure 1—figure supplement 3*), and similarly to our results in the whole-cell configuration, application of both agonists together elicited very large currents of the same magnitude as those elicited in response to 6 mM 2-APB that maximally activates channels (*Figure 1H*, *Figure 1—figure supplement 3*). These results indicate that patch excision does not disrupt sensitization of 2-APB responses by CBD, and that both agonists are capable of reaching their sites in the channel regardless of the side of the membrane to which they are applied.

## rTRPV2 channel sensitization by CBD increases channel open probability

To determine whether CBD sensitizes rTRPV2 channels to activation by 2-APB by increasing channel open probability ($P_o$), we undertook single-channel recordings. Although we were unable to obtain patches containing a single channel because the rTRPV2 channel expresses extremely well in HEK293 cells, we obtained recordings from inside-out patches containing multiple channels under conditions where the $P_o$ is very low and gating transitions of individual channels can be readily distinguished. For each patch, we recorded channel activity in the absence of agonists (i.e. control), and in the presence of 2-APB and CBD applied separately or together. All recorded data are displayed in *Figure 2A* as data blocks organized into two 3 × 4 arrays: each row of data blocks in the array contains data from a different inside-out patch (n = 6), and columns separate data obtained under different experimental conditions. For each individual data block in the array, current sweeps are stacked along the vertical axis (50 sweeps per block) and the horizontal axis within each block corresponds to the recoding duration of 500 ms per sweep. Individual data points are colored by current amplitude – see the color bar and the representative sweeps in *Figure 2C* for reference.

With the exception of a high-$P_o$ burst at 250 µM 2-APB observed in one patch, channel activity in the absence of agonists or in the presence of either 2-APB or CBD applied separately was negligible, with no clearly identifiable opening events (*Figure 2A*). In contrast, exposure of patches to 2-APB and CBD together resulted in robust channel activity in all six patches, with multiple simultaneous channel opening events observed in many of the patches (*Figure 2A and C*). Exposure to 2-APB and CBD did not elicit changes in channel activity in patches from un-transfected cells (*Figure 2B*), strongly suggesting that the increases in channel activity observed in rTRPV2-expressing cells reflect a dramatic increase in $P_o$ in rTRPV2 channels when CBD and 2-APB are simultaneously present.

To compare between data from different patches at each experimental condition, we generated all-points current amplitude histograms from each of the 24 data blocks shown in *Figure 2A*; each vertical lane in *Figure 2D* is a histogram, with current–amplitude bins on the vertical axis and a color scale to denote the logarithm of the normalized number of points per bin. For almost all patches, the histograms in control, 2-APB, or CBD have a single peak centered at 0 pA, the mean current amplitude when no channels are open. Consistent with a much greater $P_o$ in the presence of 2-APB and CBD together, the corresponding histograms all have robust peaks at larger amplitudes representing the opening of one or more channels (*Figure 2D*, right panel). We could not accurately determine single-channel current amplitudes in 2-APB or CBD because of the short duration and sparsity of openings when the agonists were applied separately. In the presence of 2-APB and CBD together, the single-channel current amplitude was centered at around 4.3 pA in five out of the six patches, with one patch exhibiting a much larger open amplitude of 8.2 pA (*Figure 2D*). In one of the patches in the presence of the two agonists, the open-channel current amplitude was initially ~4 pA but we also began observing openings with a larger current amplitude of 10.5 pA that became predominant for the rest of the experiment (*Figure 2C and D*). We are fairly confident that both amplitudes correspond

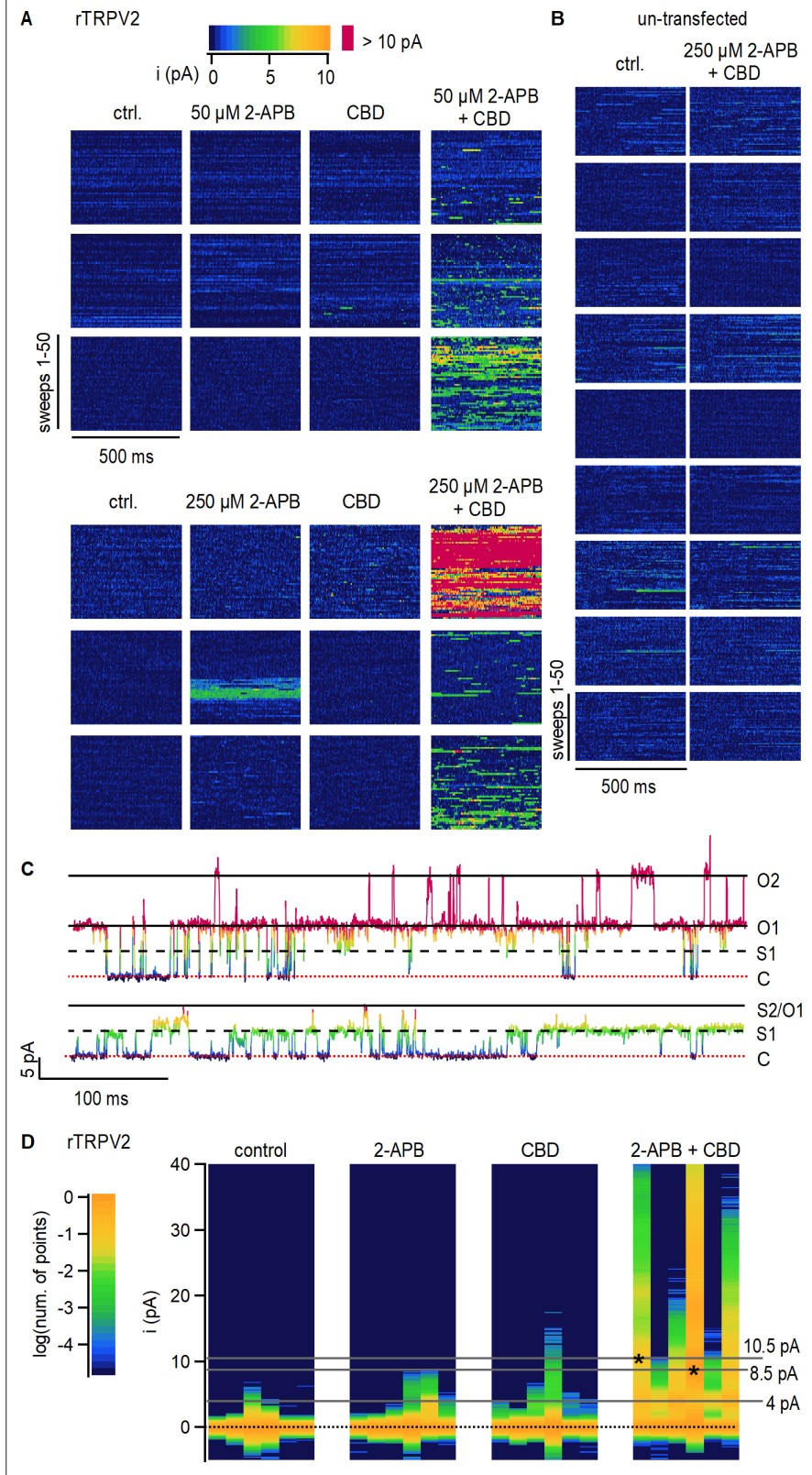

**Figure 2.** Cannabidiol (CBD) sensitization of rTRPV2 channels involves an increase in open probability. (**A**) Data obtained from six rTRPV2-expressing inside-out patches at +80 mV under conditions of low open probability. Each row of data blocks shows data from a different patch, and columns are for each of the different experimental conditions. Each data block contains 50 vertically stacked current sweeps of 500 ms duration. Data points at each

*Figure 2 continued on next page*

*Figure 2 continued*

sweep are colored by their current amplitude as indicated by the color bar at the top. (**B**) Data obtained from nine inside-out patches at +80 mV obtained from un-transfected cells, displayed as in (**A**). (**C**) Representative current traces from the same rTRPV2-expressing patch in the presence of 50 µM 2-aminoethoxydiphenyl borate (2-APB) and 10 µM CBD showing two distinct open current amplitudes (S1 and O1) and simultaneous opening of two channels (O2). Data points in each trace are colored as in (**A**). The red dotted line indicates the zero-current amplitude measured when all channels are closed. (**D**) All-points histograms for data in (**A**). Each vertical lane is a histogram, with the single-channel current amplitude bins along the y-axis and the number of points per bin shown as a log-scale color heatmap (see scale bar on the left) and normalized to the peak centered at 0 pA (black dotted line). Histograms from each patch are displayed in the same order as in (**A**). Asterisks at peak values of 8.5 or 10.5 pA indicate the patches where these single-channel amplitudes were observed.

to open rTPRV2 channels because we were able to observe multiple transitions between the two single-channel current amplitude levels when only one channel was open (*Figure 2C*). Together, these findings imply that rTRPV2 channels can undergo transitions between open states with different cation-conducting properties, as described for the closely related rTRPV1 channel (*Canul-Sánchez et al., 2018*; *Geron et al., 2018*) and the RTx-sensitive TRPV2-QM variant (*Zhang et al., 2016*), and establish that the CBD-dependent sensitization of rTRPV2 channels arises from increased open probability of channels bound to CBD and 2-APB.

## Interaction sites for CBD in the rTRPV2 channel

To further explore the mechanism by which CBD sensitizes rTRPV2 to activation by 2-APB, we set out to confirm where CBD binds (*Pumroy et al., 2019*; *Pumroy et al., 2022*) and to solve structures of rTRPV2 with both ligands. We expressed an mVenus-tagged construct of full-length rTRPV2 in mammalian cells, purified and reconstituted the protein into lipid nanodiscs using MSP1E3D1 (*Matthies et al., 2018*), and solved its structure using cryo-EM in the presence of both CBD and 2-APB (*Figure 3—figure supplements 1–4*, *Table 1*). In our initial classification and refinement, we determined the structure of rTRPV2 spanning residues F75 to S728, including the transmembrane (TM) regions along with portions of the N- and C-termini and observed reasonably well-defined density for CBD between the S5 and S6 helices (*Figure 3A*), which we termed conformation A. The overall structure of rTRPV2 in conformation A and the location where CBD binds are remarkably similar to a previously published structure of rTPRV2 with CBD bound (*Pumroy et al., 2019*), as well as a more recent structure obtained in the presence of CBD and 2-APB (*Pumroy et al., 2022*) – in each of these instances, the internal pore remains closed and the structures are likely to represent a desensitized state. Although the overall resolution of conformation A was 3.23 Å, we could not see any density corresponding to 2-APB, including regions where 2-APB has been reported to bind to either TRPV2 or TRPV3 (*Figure 3—figure supplement 5A*; *Singh et al., 2018b*; *Singh et al., 2018a*; *Zubcevic et al., 2019*; *Pumroy et al., 2022*; *Su et al., 2023*).

To identify where 2-APB binds in the presence of CBD, we undertook further focused classification of the TM region into seven classes without image alignment. Six classes were indistinguishable from conformation A, while one class shows interesting features that are distinct from conformation A which we refer to as conformation B. Although the reported resolution of conformation B was 3.32 Å, the density near the CBD binding site is much stronger compared to conformation A (*Figure 3A–C*; *Figure 3—figure supplements 1–4*, *Table 1*). In this new conformation, we observed density for CBD bound to the previously described site, similarly to conformation A. In addition, we observed a second non-protein density located nearby the S5 and S6 helices and the S4-S5 linker towards the intracellular side of the protein (*Figure 3B and C*), which represents a new potential binding site for small molecules in the rTRPV2 channel. Because this second non-protein density can be well fit by CBD in contrast to 2-APB, we have assigned this density to CBD over 2-APB based on its shape (*Figure 3C and D*), but we acknowledge that the density could represent another molecule carried over from the cells during purification. We also observed a well-defined density at the vanilloid site of a lipid molecule interacting with and likely stabilizing the second CBD molecule at this new site (*Figure 3B, E and F*). Because the headgroup density for this lipid has a conical shape that can be fit with 1-pal mitoyl-2-oleoyl-sn-glycero-3-phosphocholine (POPC) (*Figure 3E and F*), we tentatively assigned the lipid as POPC. In conformation A and earlier structures of TRPV2, the S4-S5 linker adopts a helical

**Table 1.** Cryo-EM data collection, refinement, and validation statistics.

| | Conformation A | Conformation B |
|---|---|---|
| Magnification | 105,000 | 105,000 |
| Voltage (kV) | 300 | 300 |
| Electron exposure (e-/Å$^2$) | 52 | 52 |
| Defocus range (μm) | –0.5 to –1.5 | –0.5 to –1.5 |
| Pixel size (Å) | 0.415 | 0.415 |
| Symmetry imposed | C4 | C4 |
| Initial particle images (no.) | 1,665,271 | 1,665,271 |
| Final particle images (no.) | 321,717 | 43,071 |
| Map resolution (Å) | 3.23 | 3.32 |
| FSC threshold | 0.143 | 0.143 |
| Refinement | | |
| Initial model used (PDB code) | 6U84 | 6U84 |
| Model resolution (Å) | 3.4 | 3.5 |
| FSC threshold | 0.5 | 0.5 |
| Map sharpening B-factor (Å$^2$) | –50 | –50 |
| Model composition | | |
| Non-hydrogen atoms | 17,901 | 18,313 |
| Protein residues | 2380 | 2376 |
| Ligands | 9 | 17 |
| R.m.s deviations | | |
| Bond lengths (Å) | 0.002 | 0.004 |
| Bond angles (°) | 0.455 | 0.566 |
| *B* factor(Å$^2$) | | |
| Protein | 93.49 | 78.75 |
| Ligand | 94.58 | 41.01 |
| Validation | | |
| MolProbity score | 1.29 | 1.34 |
| Clashscore | 5.45 | 6.18 |
| Poor rotamers (%) | 0 | 0 |
| Ramachandran Plot | | |
| Favored (%) | 98.84 | 98.11 |
| Allowed (%) | 1.16 | 1.89 |
| Disallowed (%) | 0 | 0 |

FSC: Fourier shell correlation.

secondary structure, whereas in conformation B, residues at the C-terminal end of the S4-S5 linker are incorporated into the S5 helix to lengthen that TM, and the remaining residues in the linker adopt an extended loop (*Figure 3B*). Importantly, the position of the vanilloid site lipid in conformation B resembles density attributed to lipid in rTRPV2 structures obtained in the presence of 2-APB together with CBD that were proposed to represent an activated state of the channel (*Pumroy et al., 2022*). In both of these instances, the position of the lipid is closer to the S4-S5 linker when compared to

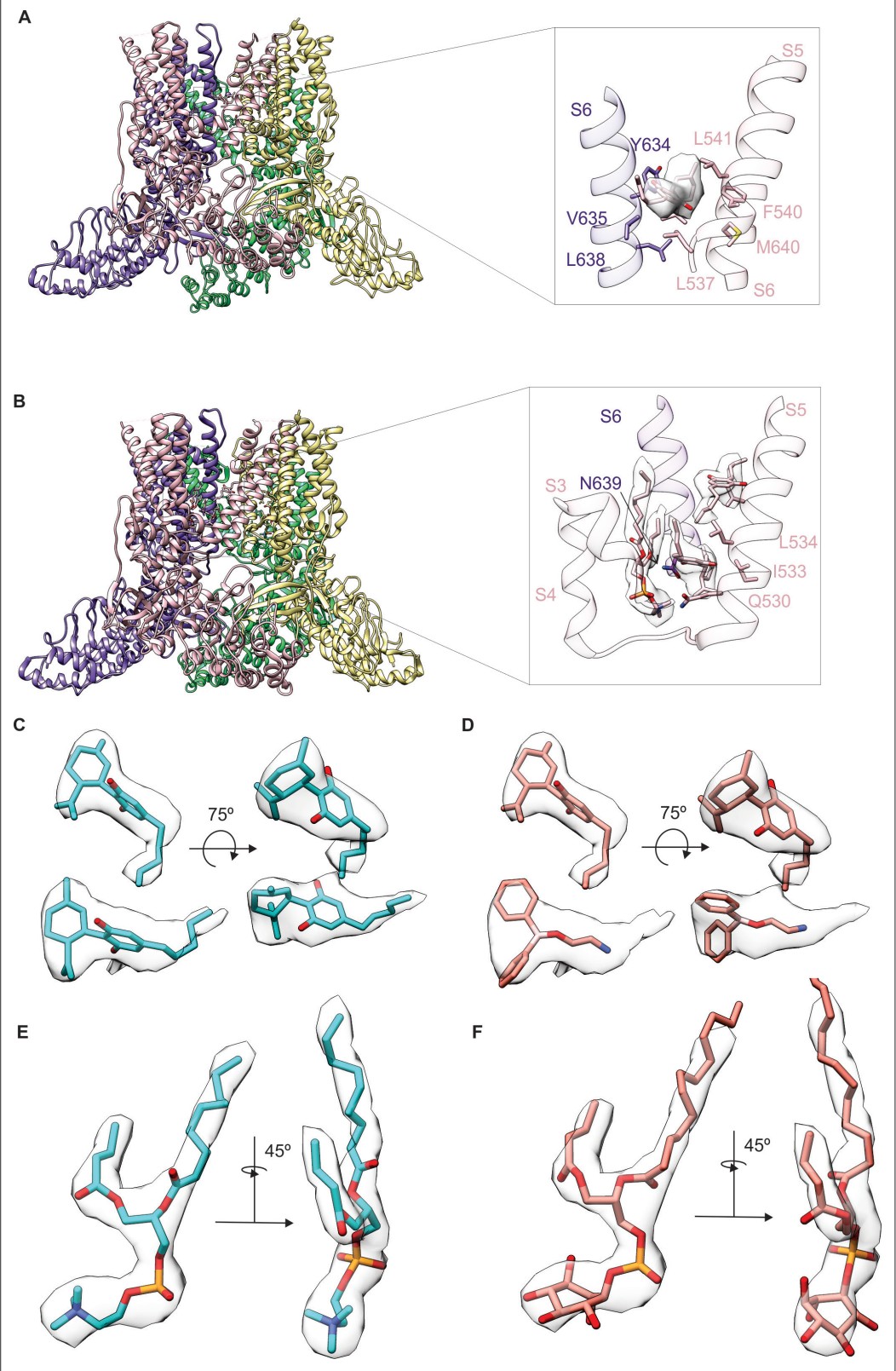

**Figure 3.** Identifying cannabidiol (CBD) binding sites in TRPV2. (**A**) Overall structure of the rTRPV2 channel in lipid nanodiscs in conformation A with one CBD molecule bound to each monomer (PDB: 8SLX). Magnified view of the single CBD binding site is shown in the right panel. Both CBD and interacting residues are presented in stick with the cryo-EM density (EMD-40582) corresponding to CBD shown as a white surface. (**B**) Overall structure of

*Figure 3 continued on next page*

*Figure 3 continued*

the rTRPV2 channel in lipid nanodiscs in conformation B with two CBD molecules bound to each monomer (PDB: 8SLY). Magnified view of the two CBD binding sites are shown in the right panel. CBD with interacting residues and lipid were presented in stick configuration and cryo-EM density (EMD-40583) corresponding to CBD and lipid are shown as a white surface. (**C**) Cryo-EM density for the two CBD binding sites in conformation B modeled using CBD and shown in two orientations. The top density corresponds to that closer to the extracellular side of the channel and the bottom towards the intracellular side. (**D**) Same cryo-EM densities shown in (**C**), but in this case modeling the more intracellular density using 2-aminoethoxydiphenyl borate (2-APB). (**E**) Cryo-EM density for the lipid near the more intracellular CBD binding site fitted with 1-palmitoyl-2-oleoyl-sn-glycero-3-phosphocholine (POPC). (**F**) Same cryo-EM density shown in (**E**), but fitted with phosphatidylinositol.

The online version of this article includes the following source data and figure supplement(s) for figure 3:

**Figure supplement 1.** Biochemistry for rTRPV2.

**Figure supplement 1—source data 1.** Uncropped SDS-PAGE of rTRPV2 in nanodisc.

**Figure supplement 2.** Data processing workflow for the cryo-EM structures of rTRPV2 in conformations A and B.

**Figure supplement 3.** Cryo-EM imaging of rTRPV2.

**Figure supplement 4.** Cryo-EM density for rTRPV2.

**Figure supplement 5.** Unassigned cryo-EM density for rTRPV2 conformations A and B in regions where density has been assigned to 2-aminoethoxydiphenyl borate (2-APB) in TRPV2 and TRPV3 channels.

**Figure supplement 6.** Structural comparison of lipids in the vanilloid pocket.

vanilloid lipid observed in other structures of TRPV2 or TRPV1 (*Figure 3—figure supplement 6*; *Zhang et al., 2021*; *Su et al., 2023*). However, as with conformation A, we see no clear density for 2-APB in conformation B, including those regions where 2-APB has been reported to bind to either TRPV2 or TRPV3 (*Figure 3—figure supplement 5B*). Although we do see extra density within the S1-S4 domain in both conformation A and B (*Figure 3—figure supplement 5A and B*), similar density can also be seen in this region for structures solved for the apo-form of TRPV2 (*Figure 3—figure supplement 5C*, left) or TRPV2 in the presence of CBD or 2-APB (*Figure 3—figure supplement 5C*, middle and right, respectively). From these results, we conclude that it remains unclear where 2-APB binds to rTRPV2, but that CBD could potentially bind to two sites within each subunit of rTRPV2, with binding to the most intracellular site involving stabilizing interactions with the lipid at the vanilloid site and a localized conformational change in the S4-S5 linker.

We note that a much larger number of particles were used to refine conformation A (321,717) compared to conformation B (43,071) (*Figure 3—figure supplement 2*), suggesting that the binding of CBD to the more extracellular site has higher occupancy compared to the more intracellular site. This difference in CBD occupancy might suggest that the affinity of CBD is higher for the more extracellular site. Another possibility is that conformation B is only transiently populated during activation or desensitization of rTRPV2. Finally, it is possible that CBD binding to the second site in conformation B requires the presence of a lipid other than the POPC that we modeled that has been carried over from purification, and the majority of particles are observed to adopt conformation A because they lost that lipid during purification. The conformational change in the S4-S5 linker observed in conformation B is interesting because this structural element plays a critical role in coupling conformational changes in the S1-S4 domain to the pore domain where the gate resides.

## CBD sensitizes rTRPV1 channels weakly and mTRPV3 channels strongly to activation by 2-APB

Rat TRPV1 (rTRPV1) and mouse TRPV3 (mTRPV3) channels share 49 and 42% amino acid sequence identity with rTRPV2, respectively, and accumulating evidence suggests that some agonists follow similar mechanisms to activate these three channels (*Yang et al., 2016*; *Zhang et al., 2016*; *Zhang et al., 2019*; *Zubcevic et al., 2018*; *Jara-Oseguera et al., 2019*; *Deng et al., 2020*; *Shimada et al., 2020*). In addition, the three channels can be activated by 2-APB (*Hu et al., 2004*), and previous reports used Ca$^{2+}$-imaging to show that CBD can also stimulate TRPV1 (*Bisogno et al., 2001*; *Ligresti et al., 2006*; *De Petrocellis et al., 2012*; *Iannotti et al., 2014*) and TRPV3 (*De Petrocellis et al., 2012*) channels with low micromolar affinity. We therefore tested whether CBD exerts a similar sensitizing effect on rTRPV1 and mTRPV3 channels. First, we confirmed that 10 µM CBD can elicit currents

in rTRPV1 or mTRPV3-expressing cells by recording current–voltage relations in the absence and presence of CBD and found that for both channels 10 µM CBD noticeably increased currents relative to control (*Figure 4A–D*).

We next probed for CBD-dependent sensitization in cells expressing rTRPV1 channels. We performed experiments where we exposed cells to 50 µM 2-APB and 10 µM CBD, first applied separately and then together, and finally we applied 10 µM capsaicin to maximally activate channels. To our surprise, we observed only moderate sensitization of the response to 2-APB in the presence of CBD (less than tenfold increase), and sensitized currents were much smaller than the maximal currents recorded in the presence of capsaicin (*Figure 4E and F*). In experiments using a higher concentration of CBD (40 µM), we obtained similar results, indicating that 10 µM CBD is enough to bind most rTRPV1 channels in the membrane, and that CBD-bound rTRPV1 channels are not strongly sensitized to activation by 2-APB, contrary to our observations with rTRPV2 channels.

We then tested the sensitizing effect of CBD on mTRPV3-expressing cells. We used a concentration of 2-APB that maximally activates channels (60 µM), albeit with extremely slow kinetics due to the large energy barrier associated with the activation of this channel (*Liu et al., 2011*). Following activation, there is an even larger energy barrier in the reverse direction (i.e. back towards the initial closed state before exposure to 2-APB), resulting in the irreversible sensitization of channels. Sensitized and non-sensitized channels therefore open from different closed states upon stimulation by 2-APB, and the energy barrier for activation is much smaller for sensitized channels such that the kinetics of activation by 2-APB become much faster after sensitization (*Liu et al., 2011*; *Zubcevic et al., 2019*). We found that a brief 10 s exposure to 2-APB resulted in a slow and minimal increase in current, whereas exposing the same cell to 2-APB together with CBD resulted in maximal current activation with a time constant <10 s (*Figure 4G and K*). In contrast, in experiments where cells were repeatedly exposed to 60 µM 2-APB in the absence of CBD, currents elicited by the agonist took several minutes to reach a steady state at maximal activation (*Figure 4H and K*), consistent with previous observations (*Liu et al., 2011*; *Zubcevic et al., 2019*; *Deng et al., 2020*). We noticed that over the course of an experiment the activation kinetics of currents elicited by 2-APB changed with each exposure to the agonist. To evaluate this effect, for each exposure to agonist we fitted a bi-exponential function of time to the current time course (*Figure 4H*, red curves), and averaged the resulting slow (yellow) and fast (magenta) time constants across experiments at equivalent timepoints where exposure to agonist was initiated (*Figure 4K*, left panel, yellow and magenta symbols). During the first exposure to 2-APB, current activation was mono-exponential and slow – likely because at the beginning of the experiment there are no sensitized channels, and so activation kinetics are dominated by the slow activation of non-sensitized channels. Currents followed a bi-exponential time course during the second and all subsequent exposures to agonist. The slow time constants were similar to what we measured during the first stimulation, whereas the fast time constants were similar to what we obtained from mono-exponential fits to currents elicited by 2-APB in combination with CBD (*Figure 4K*, right panel, black circles). We interpret this fast component as originating from the fraction of channels in the cell that have become sensitized by 2-APB. We think that the slow time constant reflects the slow process of sensitization by 2-APB. Consistently, the slow time constants obtained from fits to individual 2-APB exposures are similar to those obtained from a mono-exponential fit to the time course spanning the entire experiment (*Figure 4K*, right panel, blue circles). Together these data indicate that channels activate fast when sensitized by either 2-APB or 2-APB + CBD, but the latter experimental condition allows channels to reach the sensitized state much more rapidly than 2-APB binding alone.

Our results so far indicate that CBD can sensitize mTRPV3 channels much more rapidly than 2-APB at 60 µM. We next tested whether 2-APB at a much higher concentration (3 mM) would sensitize channels much more rapidly. We first found that pre-stimulation of cells with 3 mM 2-APB had negligible effect on the kinetics of activation by 60 µM 2-APB applied together with CBD, confirming that CBD binding rapidly sensitizes all channels in the recorded cell membrane to activation by 2-APB (*Figure 4I and K*). In contrast, we found that currents elicited by 60 µM 2-APB remained bi-exponential and continued to slowly increase upon repeated exposures after pre-stimulation with 3 mM 2-APB for >1.5 min (*Figure 4J and K*). These results establish that CBD is a much weaker agonist than 2-APB for the mTRPV3 channel, but has a more potent sensitizing effect, similarly to our results with rTRPV2 channels. The absence of a strong sensitizing effect of CBD in rTRPV1 channels points to

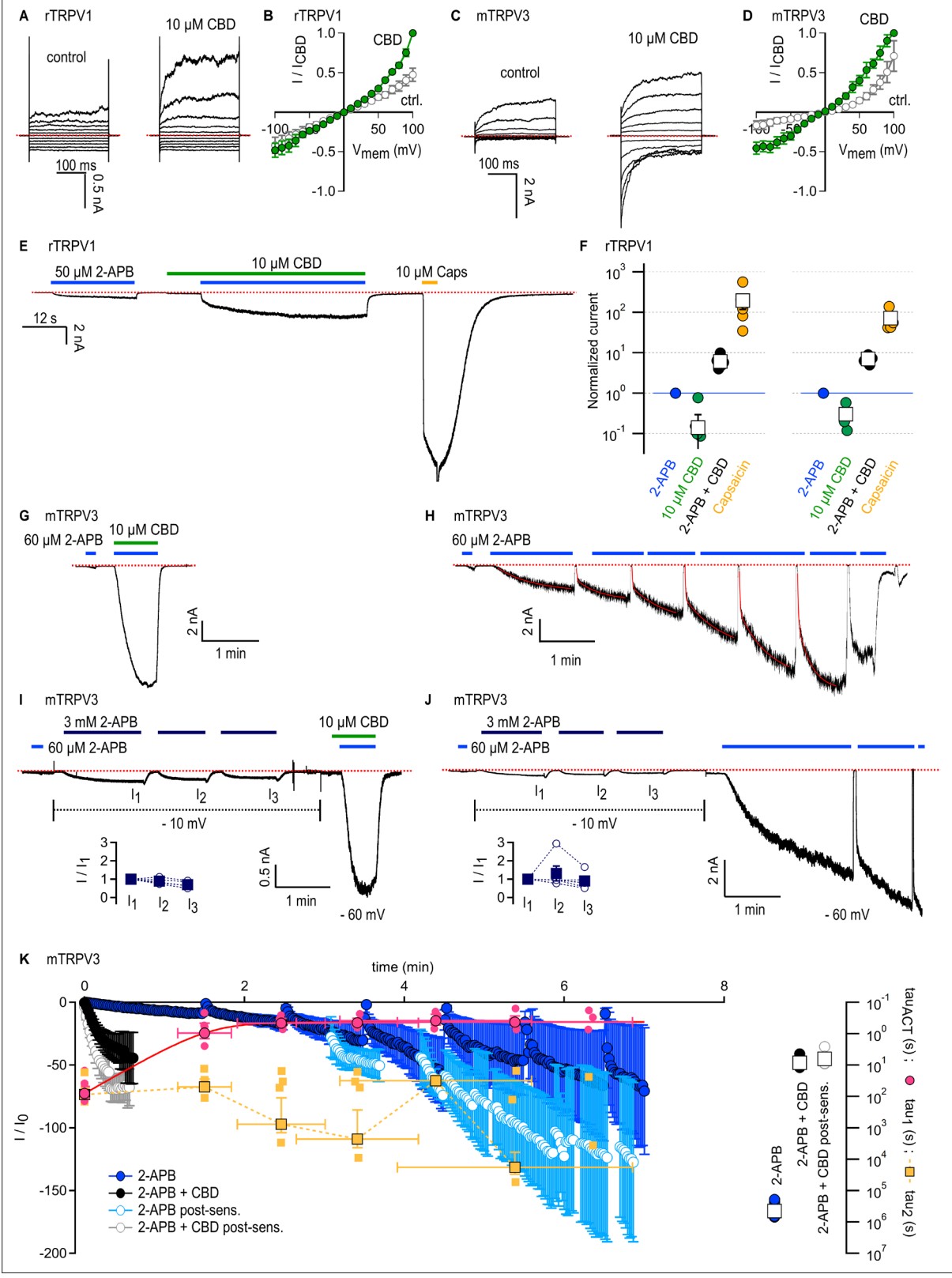

**Figure 4.** Cannabidiol (CBD) sensitizes rTRPV1 channels weakly and mTRPV3 channels strongly to activation by 2-aminoethoxydiphenyl borate (2-APB). (**A**) Representative whole-cell current families obtained from rTRPV1 expressing cells in control or in the presence of 10 μM CBD. Currents were elicited by voltage steps from –100 to +100 mV. The red dotted line denotes the zero-current level. (**B**) Steady-state current–voltage relations obtained from data as in (**A**) and normalized to the steady-state current magnitude at +100 mV in CBD. Data is shown as mean ± SEM (n = 6). (**C**) Representative whole-

*Figure 4 continued on next page*

*Figure 4 continued*

cell current families from mTRPV3-expressing cells obtained as in (**A**). (**D**) Steady state mTRPV3 channel current–voltage relations obtained from data as in (**C**) (mean ± SEM, n = 7). (**E**) Representative whole-cell recording at –80 mV obtained from an rTRPV1-expressing cell. (**F**) Group data from experiments as in (**E**) obtained using 10 or 40 μM CBD, leak-subtracted and normalized to the current magnitude measured at steady state during the first stimulation with 50 μM 2-APB, denoted by the horizontal blue line. Data are shown as mean ± SEM (squares, 10 μM CBD, n = 6; 40 μM CBD, n = 5) or as individual cells (circles). (**G**) Representative whole-cell gap-free recording at –60 mV obtained from a cell expressing mTRPV3 channels stimulated with 2-APB and CBD. The first exposure to 2-APB alone (10 s duration) was used for normalization. (**H**) Representative experiment showing repeated stimulation with 2-APB. The first exposure to 2-APB (10 s duration) was used for normalization. Red curves are fits to a double-exponential function of time, with resulting fast (tau$_1$) and slow (tau$_2$) time constants in magenta and yellow superposed to the mean time-course in (**K**). (**I, J**) Representative experiments at – 60 mV for sensitized responses to (**I**) 2-APB and CBD or (**J**) 2-APB alone measured after three exposures to 3 mM 2-APB applied at –10 mV to preserve patch integrity (dotted black lines). The first exposure to 60 μM 2-APB (10 s duration) was used for normalization. The inset shows current amplitudes at each stimulation with 3 mM 2-APB (filled squares, mean ± SEM; empty circles, data from individual cells, n = 5). (**K**) Mean time courses for mTRPV3 channel activation by 60 μM 2-APB in the absence (blue symbols) or presence of 10 μM CBD (black symbols), measured in experiments as in (**G**) and (**H**), as well as sensitized responses to 2-APB (light blue symbols) or 2-APB + CBD (gray symbols) measured as in (**I**) and (**J**). The time course for the sensitized response to 2-APB alone does not start at t = 0 to account for the sensitizing exposures to 3 mM 2-APB. Data are shown as mean ± SEM (n = 5 for each condition). Data were fit to a mono-exponential function of time, with time constants shown on the right inset as mean ± SEM (open squares) or data from individual cells (circles). The yellow and magenta symbols are the time constants obtained from double-exponential fits to each stimulation by 2-APB over the course of experiments as in (**H**). Data is shown as mean ± SEM (symbols with black outline) as well as fits from each individual cell (smaller symbols without outline). The magenta curve is a fit to a mono-exponential function.

The online version of this article includes the following source data for figure 4:

**Source data 1.** Excel file with group data from electrophysiological recordings shown in *Figure 4B and D*.

**Source data 2.** Excel file with group data from electrophysiological recordings shown in *Figure 4F*.

**Source data 3.** Excel file with group data from electrophysiological recordings show in *Figure 4I–K*.

a key energetic difference regarding activation of rTRPV1 channels by 2-APB relative to rTRPV2 and mTRPV3 channels.

## Structural determinants of sensitization strength by CBD in rTRPV2 and rTRPV1 channels

We next sought to identify the structural determinants for the strong sensitization by CBD observed in the rTRPV2 channel relative to the weak sensitization observed in the rTRPV1 channel. The CBD site first identified in the full-length rTRPV2 channel (*Pumroy et al., 2019*) shows high conservation at the amino acid sequence level between the TRPV1, TRPV2, and TRPV3 channels (*Figure 5*, *Figure 5— figure supplement 1*). Under the assumption that sensitization in CBD-bound rTRPV2 and mTRPV3 channels follows a similar mechanism, we began by inspecting the amino acid sequence alignment between the three channels and identified three residues in the S5 helix in the pore domain and the S4-S5 linker that are identical in rTRPV2 and mTRPV3 channels and different in rTRPV1 channels, and are also located very close to the CBD site in conformation A that is also supported by previous studies (*Pumroy et al., 2019*; *Pumroy et al., 2022*; *Figure 5*, *Figure 5—figure supplement 1*). We generated six mutant channels where we individually swapped each of the non-conserved residues between the rTRPV2 (V532M, L538C, L541M) and rTRPV1 channels (M572V, C578L, M581L). For each channel, we also generated a double (rTRPV2 L538C+L541M; rTRPV1-C578L+M581L) and a triple mutant (rTRPV2-V532M+L538C+L541M; rTRPV1-M572V+C578L+M581L). Our hypothesis was that if those amino acid side chain differences determine the strong or weak sensitization to 2-APB in CBD-bound rTRPV2 or rTRPV1 channels, respectively, then these introduced mutations should weaken sensitization in rTRPV2 channels and strengthen it in rTRPV1 channels.

We began by measuring the 2-APB concentration-response relations for all constructs (*Figure 6— figure supplement 1A–F*). Notably, the apparent affinity for 2-APB was nearly 10-fold higher in WT rTRPV1 than in the rTRPV2 channel (EC$_{50}$ = 0.295 ± 0.22 mM in WT rTRPV1 vs. EC$_{50}$ = 1.88 ± 0.15 mM in WT rTRPV2), and activation was somewhat less cooperative (*Figure 6—figure supplement 1C and F*). The L541M mutation closest to the CBD binding site in the rTRPV2 channel conformation A structure resulted in a small but appreciable increase in apparent affinity for 2-APB, which became more pronounced in the double mutant (rTRPV2 L538C+L541M EC$_{50}$ = 0.97 ± 0.18 mM; *Figure 6—figure supplement 1A and B*). The corresponding mutations in TRPV1 had the opposite trend, with a small yet appreciable decrease in apparent affinity in the single mutants (C578L and M581L) and a slightly

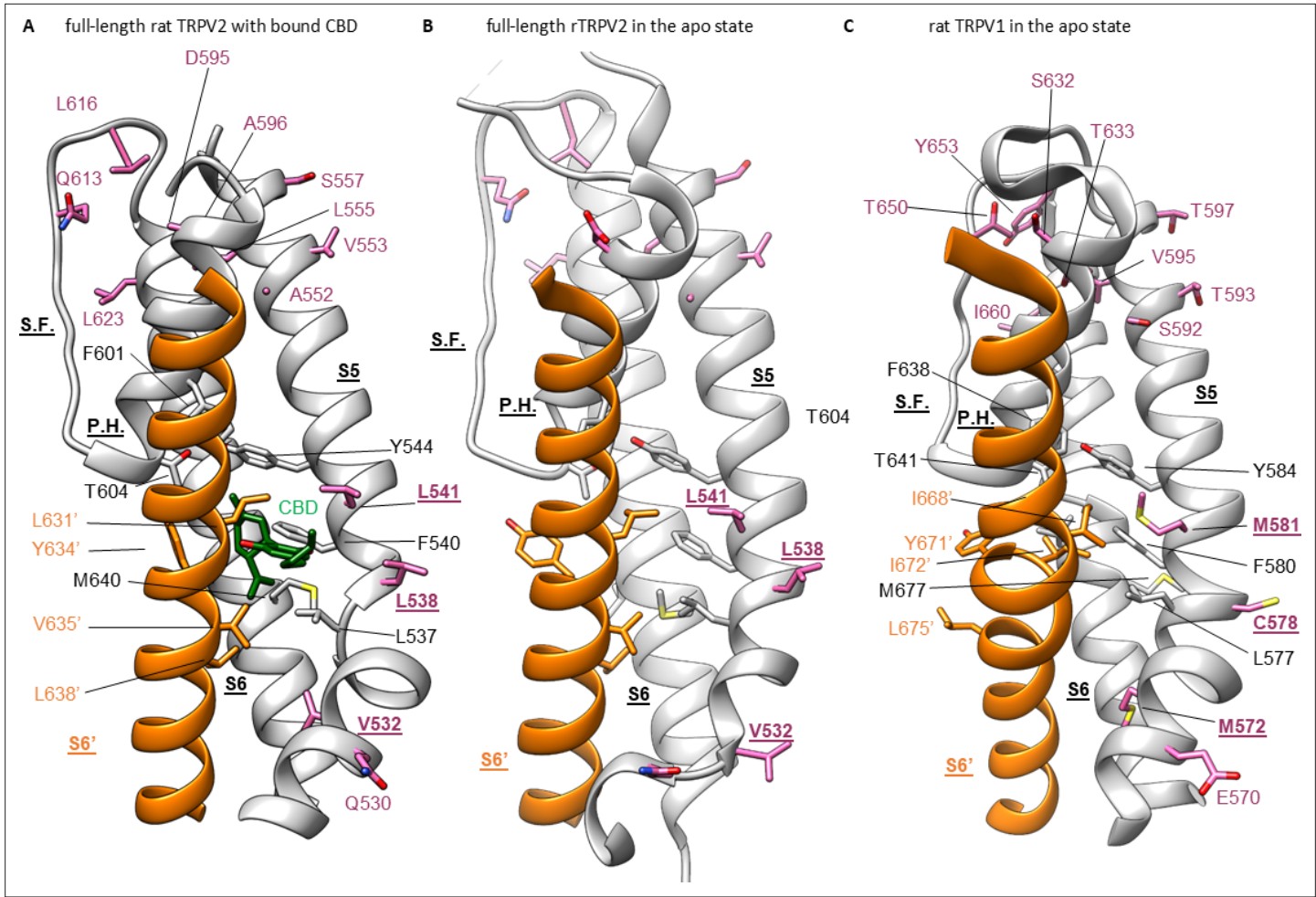

**Figure 5.** Cannabidiol (CBD) binding site in TRPV2 channels and its conservation in TRPV1 channels. (**A**) Structure of full-length rTRPV2 in nanodiscs with CBD bound in conformation A (with one CBD per subunit; PDB: 8SLX). The S6 helix from one subunit is shown in orange, and in the adjacent subunit the S5 and S6 helices, the selectivity filter (S.F.), the pore-helix (P.H.), and the S4-S5 linker helix are shown in gray. Residues near the CBD site are shown in stick representation, and the bound CBD molecule in green. Residues that are similar in rTRPV2 and mTRPV3 channels but different in rTRPV1 channels are shown in purple. (**B**) Structure of apo full-length rTRPV2 in nanodiscs (PDB: 6U84) (**Pumroy et al., 2019**). Same color coding as in (**A**). (**C**) Structure of apo rTRPV1 (PDB: 5IRZ) (**Gao et al., 2016b**) depicting the CBD binding region from rTRPV2. Same color coding as in (**A**) and (**B**).

The online version of this article includes the following figure supplement(s) for figure 5:

**Figure supplement 1.** Sequence alignment of rodent and human TRPV1-3 orthologues.

larger decrease in the double mutant (rTRPV1 C578L+M581L EC$_{50}$ = 1.15 ± 0.14 mM; *Figure 6— figure supplement 1D and E*). Mutation at position V532 in rTRPV2 had the most profound effect: the single (V532M) and the triple (V532M+L538C+L541M) mutants were unresponsive to 2-APB or CBD, indicating that V532 is critical for rTRPV2 channel operation, potentially contributing to folding, surface expression, or gating. Consistent with a critical role for this site in gating, we found that mutation M572V in rTRPV1 impacted agonist sensitivity the most: the single mutation drastically reduced the apparent affinity for 2-APB to be even lower than in WT rTRPV2 (rTRPV1 M572V EC$_{50}$ = 3.32 ± 0.2 mM), and for the triple mutant (rTRPV1 M572V+C578L+M581L) we were unable to determine an EC$_{50}$ because even the highest attainable 2-APB concentrations in solution were insufficient to maximally activate channels (*Figure 6—figure supplement 1D and E*). Interestingly, for all rTRPV1 channel mutants we found a much stronger reduction in the apparent affinity for 2-APB than for capsaicin, and for the two constructs containing the M572V, the efficacy of activation by 2-APB was also reduced compared to capsaicin (*Figure 6—figure supplement 1G and H*).

Although none of the side chains we mutated are reported to contact 2-APB when bound to any of its proposed sites in TRPV2 (*Pumroy et al., 2022*; *Su et al., 2023*), we observed altered sensitivity

to this agonist in most mutant channels. This observation is consistent with our hypothesis that the region containing the CBD site in conformation A exerts allosteric control over the response to 2-APB by both rTRPV2 and rTRPV1 channels. We therefore proceeded to test whether sequence differences between the rTRPV2 and rTRPV1 channels at any of these three sites are responsible for the differences in sensitization strength between the two channels. To test our hypothesis, we performed experiments similar to those in *Figure 1A* and assessed sensitization at three different concentrations of CBD to ensure that for each mutant we measured sensitization at full channel occupancy by this ligand (*Figure 6A–F*, *Figure 6—figure supplement 2A–E*). To account for differences in sensitivity to 2-APB in each mutant when assessing sensitization, we related the current elicited by a sub-activating concentration of 2-APB ($I_{2APB}$) to a maximally activating concentration of 2-APB ($I_{2APB,max}$), given by the current ratio $I_{2APB,max}/I_{2APB}$, as well as to the sensitized response to 2-APB in the presence of the highest concentration of CBD ($I_{CBD+2APB}$), given by the ratio $I_{CBD+2APB}/I_{2APB}$. For experiments with WT rTRPV2, where CBD-sensitized responses to 2-APB reached maximal channel activation, the two ratios yielded similar values and when graphed against each other, data localized close to the diagonal given by $I_{2APB,max}/I_{2APB} = I_{CBD+2APB}/I_{2APB}$ (*Figure 6G*). For the WT rTRPV1 channel, which had weaker sensitization, data on the same plot localized further down from the diagonal.

For the rTRPV2 mutants, we did not detect any notable differences relative to WT channels except for the double mutant rTRPV2 L538C+L541M, where the sensitized response at 10 µM CBD was less than maximal (*Figure 6C*, *Figure 6—figure supplement 2E*), suggesting that the combined mutations reduced apparent affinity for CBD. For rTRPV1, mutations C578L and M581L appeared to slightly increase sensitized responses, which became more marked for the double mutant, with data from the single and double mutants falling closer than WT to the diagonal on *Figure 6G*. However, sensitized responses in these rTRPV1 channel mutants remained lower than maximal activation, and weaker than for rTRPV2 channels. Notably, channels carrying the single mutation M572V lacked sensitization altogether (i.e. $I_{CBD+2APB}/I_{2APB} < 1$, *Figure 6E and G*), and in the triple mutant M572V+C578L+M581L sensitization was restored to levels similar to WT rTRPV1 (*Figure 6F and G*). Interestingly, the M572V mutation brought down the $EC_{50}$ for 2-APB close to that of WT rTRPV2 without enhancing sensitization by CBD, establishing that apparent affinity for 2-APB and sensitization strength by CBD are not obligatorily coupled. Together, these results support the idea that the region containing the CBD site observed in conformation A of the rTRPV2 channel exerts allosteric control on the response to 2-APB in both rTRPV1 and rTRPV2 channels, and that position V532 in rTRPV2 and M572 in rTRPV1 has a prominent allosteric role in this mechanism. However, our results also showed that the difference in sensitization strength that we observed between the rTRPV2 and rTRPV1 channels is not determined by amino acid sequence differences near the CBD binding site in rTRPV2 conformation A.

The second tentative CBD site that we observed in conformation B could be potentially involved in determining sensitization strength by CBD in the rTRPV2 and rTRPV1 channels. Importantly, the side chain at position Q530 localizes near to the bound CBD molecule (*Figure 3B*) and is not conserved in rTRPV1. We therefore tested whether mutations to alanine and glycine that would eliminate a potential interaction with that side chain in rTRPV2 would weaken sensitization by CBD. However, we found that both mutant channels had strong sensitization comparable to that of WT rTRPV2 channels (*Figure 6—figure supplement 3*), leaving the relevance of the second CBD site for mediating sensitization in TRPV2 an open question.

We and others have shown that residues in the pore-domain can dramatically tune sensitivity to activators in TRPV1-3 channels (*Jordt et al., 2000*; *Grandl et al., 2008*; *Grandl et al., 2010*; *Jara-Oseguera et al., 2016*; *Zhang et al., 2019*). We therefore proceeded to interrogate whether amino acid sequence differences between the pore domains of rTRPV2 and rTRPV1 channels could be responsible for their differences in sensitization strength by CBD. We found eight additional positions in the pore domain that are conserved in rTRPV2 and mTRPV3 channels but not in rTRPV1 channels (*Figure 5*, *Figure 5—figure supplement 1*) – A552, V553, L555, and S557 in the S5 helix of rTRPV2 (S592, T593, V595, T597 in rTRPV1), D595 and A596 in the pore helix of rTRPV2 (S632, T633), Q613 above the selectivity filter in rTRPV2 (T650 in rTRPV1), and L623 in the S6 helix of rTRPV2 (I660 in rTRPV1). We cumulatively substituted each of these positions in rTRPV1 with the amino acids present in rTRPV2, together with four substitutions near the CBD sites (E570Q near CBD site 2, equivalent to Q530 in rTRPV2, and CBD site 1 residues M572V, C578L, M581L in TRPV1), and tested whether CBD had a stronger sensitizing effect in the resulting rTRPV1-12M channel. The sensitivity to 2-APB

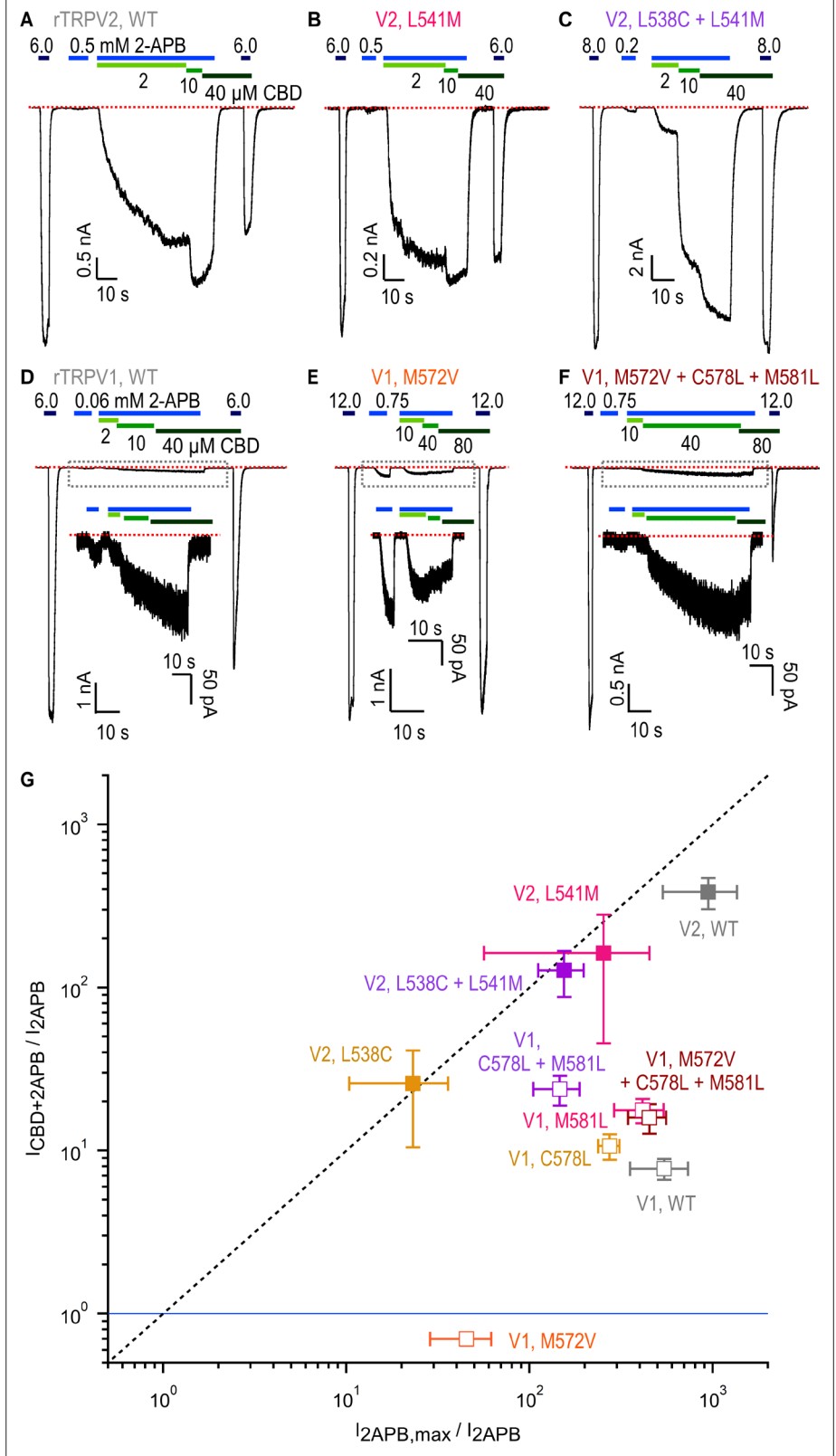

**Figure 6.** Role of non-conserved residues at the cannabidiol (CBD) site in determining rTRPV1 and rTRPV2 channel sensitivity to CBD and 2-aminoethoxydiphenyl borate (2-APB). (**A–C**) Representative gap-free whole-cell recordings at –80 mV obtained from cells expressing WT or mutant rTRPV2 channels. Horizontal bars denote the time of exposure to 2-APB (blue bars, mM concentrations) or CBD (green bars, µM concentrations). The red dotted line

*Figure 6 continued on next page*

*Figure 6 continued*

denotes the zero-current level. (**D–F**) Representative gap-free whole-cell recordings at –80 mV obtained from cells expressing WT or mutant rTRPV1 channels. Insets show a magnified view of the portion of the time courses contained within the dotted rectangles. (**G**) Leak-subtracted group data for WT and mutant rTRPV1 (empty symbols) and rTRPV2 (filled symbols) channels obtained from experiments as in (**A–F**) relating the response to a low concentration of 2-APB ($I_{2APB}$) to either the maximal response to the same agonist ($I_{2APB,max}/I_{2APB}$, horizontal axis) or the sensitized response measured at the highest concentration of CBD in each experiment ($I_{CBD+2APB}/I_{2APB}$, vertical axis). The blue horizontal line denotes $I_{CBD+2APB}/I_{2APB} = 1$ (i.e. no sensitized response), and the dashed vertical line denotes $I_{CBD+2APB}/I_{2APB} = I_{2APB,max}/I_{2APB}$. Data are shown as mean ± SEM (WT rTRPV2, n = 4; L538C, n = 4; L541M, n = 4; L538C+L541M, n = 4; WT rTRPV1, n = 5; M572V, n = 5; C578L, n = 5; M581L, n = 5; C578L+M581L, n = 5; M572V+C578L+M581L, n = 6). All group data for measurements in experiments as in (**A–F**) are shown in *Figure 6—figure supplement 2*.

The online version of this article includes the following source data and figure supplement(s) for figure 6:

**Figure supplement 1.** Effects of mutations at non-conserved residues at the cannabidiol (CBD) site on rTRPV1 and rTRPV2 channel sensitivity to agonists.

**Figure supplement 1—source data 1.** Excel file with 2-aminoethoxydiphenyl borate (2-APB) dose–response relation data for rTRPV1 and rTRPV2 mutants.

**Figure supplement 1—source data 2.** Excel file with group data from dose–response relations for capsaicin.

**Figure supplement 2.** CBD-dependent sensitization in WT and mutant rTRPV1 and rTRPV2 channels.

**Figure supplement 2—source data 1.** Excel file with group data from electrophysiological recordings of cannabidiol (CBD)-dependent sensitization for WT and mutant rTRPV1 and rTRPV2 channels.

**Figure supplement 3.** Mutations near the rTRPV2 channel cannabidiol (CBD) binding site 2 do not affect sensitization strength.

**Figure supplement 3—source data 1.** Excel file with group data from electrophysiological recordings of cannabidiol (CBD)-dependent sensitization for rTRPV2 Q530A and Q530G channels.

---

was highly disrupted in rTRPV2-12M channels (*Figure 7A*), with a concentration of 12 mM 2-APB producing a similar extent of activation as 50 µM 2-APB in WT rTRPV1 channels relative to 10 µM capsaicin. (*Figure 4E and F*). In contrast to the heavily impaired sensitivity to 2-APB, sensitivity to CBD, and the magnitude of the sensitization caused by CBD appeared to be minimally affected relative to WT rTRPV1 (*Figure 7A and C*). Capsaicin sensitivity was also seemingly less affected, as we observed robust responses to a concentration of 10 µM that were much greater than those elicited by the highest concentration of 2-APB that we tested. Finally, we introduced two additional substitutions at the extracellular S6 loop (Y653L and D654R) of rTRPV1-12M channels, which are not conserved between rTRPV2, mTRPV3, and TRPV1 channels (*Figure 5—figure supplement 1*), and found that the resulting channels, rTRPV1-14M, behaved very similarly to rTRPV1-12M channels (*Figure 7B and C*). Together, our results establish that differences in the amino acid sequence of the pore domain between the rTRPV2 and rTRPV1 channels are not responsible for the different sensitization strength caused by CBD binding in each of these two channels. These results also indicate that the determinants for sensitization strength must be located at other sites of the protein further from the CBD binding site(s).

## Discussion

The goal of this study was to characterize the actions of the cannabinoid CBD on the TRPV2 channel. Despite the low efficacy of CBD relative to 2-APB as an agonist of rTRPV2, rTRPV1, and mTRPV3 channels, CBD sensitized all three channels to subsequent activation by 2-APB. However, sensitization in rTRPV2 and mTRPV3 channels was orders of magnitude stronger than in rTRPV1 channels. Our cryo-EM structural data for the full-length rTRPV2 in the presence of CBD and 2-APB confirmed the location of the CBD binding site (*Pumroy et al., 2019*; *Pumroy et al., 2022*) at the interface between the pore domain, the membrane and the S4-S5 linker helix (*Figure 3*), and identified a second location where CBD or other small molecules could possibly bind. The regions where CBD binds in rTRPV2 channels are highly conserved between TRPV1, TRPV2, and TRPV3 channels (*Figure 5*, *Figure 5—figure supplement 1*), and our mutagenesis results with rTRPV1 and rTRPV2 channels establish that the few residues that are not conserved between these channels are not determinant for the strength

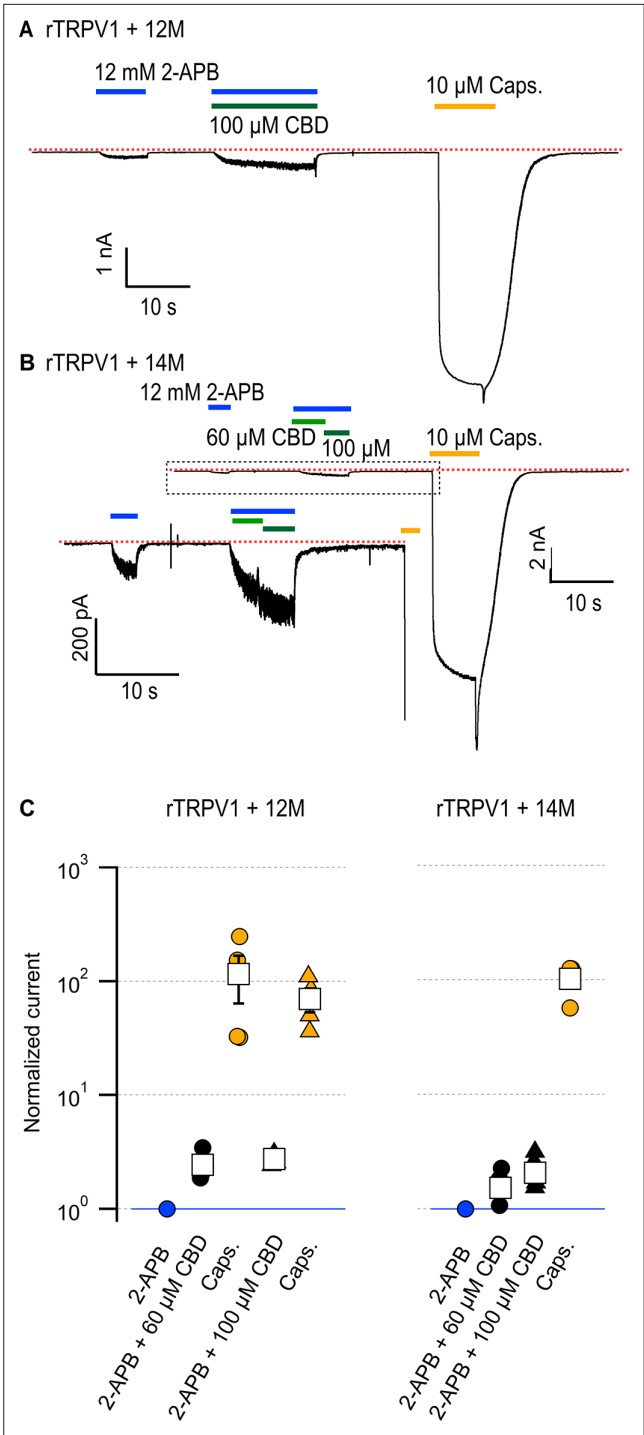

**Figure 7.** Cumulative substitutions at non-conserved positions in the rTRPV1 pore do not enhance sensitization. Representative time courses for (**A**) rTRPV1-12M channels (E570Q + M572V + C578L + M581L + S592A + T593V + V595L + T597S + S632D + T633A + T650Q + I660L) or (**B**) rTRPV1-14M channels (E570Q + M572V + C578L + M581L + S592A + T593V + V595L + T597S + S632D + T633A + Y653L + D654R + T650Q + I660L) displaying sensitization by cannabidiol (CBD) of the response to 12 mM 2-aminoethoxydiphenyl borate (2-APB). The red dotted line indicates the zero-current level. The inset in (**B**) is a magnification of the region within the dotted rectangle. Concentrations of 60 or 100 µM CBD were tested in separate experiments for rTRPV1-12M and in the same experiment for rTRPV1-14M. (**C**) Group data for the experiments in (**A**) and (**B**), showing leak-subtracted current responses to each of the stimuli, normalized to the current magnitude in the presence of 12 mM 2-APB, denoted

*Figure 7 continued on next page*

*Figure 7 continued*

by the horizontal blue line. Data points from individual experiments are shown as circles. Data are shown as mean ± SEM (n = 4 for rTRPV1-12M and 60 or 100 µM CBD; n = 5 for rTRPV1-14M).

The online version of this article includes the following source data for figure 7:

**Source data 1.** Excel file with group data from electrophysiological recordings of cannabidiol (CBD)-dependent sensitization shown in *Figure 7*.

of the sensitization caused by CBD binding. Based on these findings, we propose that CBD interacts at similar locations in TRPV1 and TRPV2 channels, and speculate that this may also apply to TRPV3 channels. In contrast to the CBD site, the 2-APB binding sites proposed based on observations in TRPV3 (*Singh et al., 2018b*; *Zubcevic et al., 2019*) and TRPV2 (*Pumroy et al., 2022*; *Su et al., 2023*) channel structures lack sequence conservation, suggesting that the energetics of activation by 2-APB could be very different in TRPV1, TRPV2, and TRPV3 channels, and could even involve distinct regions in each channel. Importantly, we observed no evidence for 2-APB-associated densities in our cryo-EM maps (*Figure 3—figure supplement 5*), calling for experimental clarification regarding the site or sites where 2-APB interacts with each of these channels.

We hypothesize that binding of CBD in rTRPV2 channels favors rearrangements on multiple sites in the protein that contribute energetically to activation by 2-APB without directly influencing the open-to-closed equilibrium because CBD is such a low-efficacy agonist. Because binding of CBD sensitizes responses to 2-APB in rTRPV1 and rTRPV2 channels, albeit with different strength, it can be predicted that mutations at the site where CBD binds would significantly affect sensitivity to 2-APB by means of allosteric coupling. Indeed, we found that each of the positions we substituted had a minor influence on the sensitivity of rTRPV1 and rTRPV2 channels to CBD and a stronger influence on the sensitivity to 2-APB; further, each of the rTRPV1 channel mutants we examined affected sensitivity to 2-APB more strongly than to capsaicin (*Figure 6—figure supplement 1*). Of note, rTRPV1 channels have an increased apparent affinity for 2-APB relative to rTRPV2 channels, with an $EC_{50}$ ~ 10-fold lower than that of rTRPV2 channels (*Figure 6—figure supplement 1A and B*). It could thus be hypothesized that the absence of strong sensitization in rTRPV1 channels results from these channels already existing in a sensitized conformation (*Zubcevic et al., 2019*). Our mutagenesis results, however, show that the apparent affinity for 2-APB and sensitization by CBD are not strictly coupled in rTRPV1 channels: mutant M572V had an $EC_{50}$ for 2-APB that is similar in magnitude to WT rTRPV2 channels, and yet sensitization by CBD was completely absent, whereas the apparent 2-APB affinity for the triple mutant rTRPV1-M572V+C578L+M581L was so low that an $EC_{50}$ could not be determined, and yet responses to 2-APB were sensitized by CBD to the same extent as in WT rTRPV1 channels (*Figure 6E–G*). Observations like this suggest a framework of TRPV channel function where the ensembles of conformational channel states stabilized by distinct agonists and allosteric modulators differ from each other at multiple significant locations throughout the protein, and that these differences, which could involve relatively subtle structural rearrangements, ultimately determine how the ensemble will respond to further challenges.

There appear to be many independent mechanisms through which TRPV2 channels can become sensitized to activation by 2-APB. Our results clearly indicate that sensitization caused by 2-APB itself (*Liu and Qin, 2016*) is over an order of magnitude weaker than that caused by CBD (*Figure 1G and H*). This large difference suggests that the ensemble of conformations adopted by the rTRPV2 channel when bound to CBD must have energetically significant differences from the set of conformations that predominate after channels become sensitized by a strong stimulation with 2-APB. Sensitization by extreme heat increases the apparent affinity of rTRPV2 channels for 2-APB to a similar extent as CBD does, and also results in noticeable channel activity at ~40°C (*Liu and Qin, 2016*). We did not detect rTPRV2 channel activity at ~40°C in the presence of CBD (*Figure 1—figure supplement 2*), indicating that CBD binding is not strongly coupled to heat activation, and suggesting that the structural ensemble of heat-sensitized channels also bears significant differences at the energetic level from that of CBD-bound channels at temperatures below 40°C. Interestingly, oxidative modification of two methionine residues in rTRPV2 channels strongly sensitized their responses to both heat and 2-APB (*Fricke et al., 2019*), indicating that methionine oxidation is less specific in its action than CBD, and suggesting that the underlying mechanism involves regions central to channel activation regardless of the stimulus.

In the case of mTRPV3 channels, we found that sensitization by 2-APB occurred much slower than by CBD, even when very high concentrations (3 mM) of 2-APB were used to pre-sensitize channels (*Figure 4G–K*). However, we also observed that current activation kinetics in the presence of CBD + 2-APB were similar to those after 2-APB-dependent sensitization (*Figure 4G, H and K*), suggesting that CBD-bound and 2-APB-sensitized apo mTRPV3 channels could adopt similar structural ensembles. It is unclear why CBD is so effective at sensitizing mTRPV3 channels without robustly activating them, whereas 2-APB can maximally activate channels but only after a slow sensitizing process – perhaps the CBD site is widely accessible in non-sensitized, apo mTRPV3 channels, whereas accessibility for 2-APB at the site that causes sensitization requires additional conformational rearrangements in the protein (and possibly the associated lipids; *Su et al., 2023*). The binding of CBD could rapidly sensitize channels by promoting these latter rearrangements and allowing 2-APB to access its sensitizing site. Importantly, the remarkably potent effect of CBD on mTRPV3 channel activation establishes it as a promising experimental manipulation that can help understand the unique kinetic features that characterize TRPV3 channel activation.

It will be interesting to test whether other molecules in addition to CBD can have similar effects on the sensitivity of TRPV2 and TRPV3 channels to 2-APB and possibly other agonists, especially because both ion channels have few known activators, endogenous or synthetic. Future drug or activator screens for TRPV2 and TRPV3 channels could benefit from combining test drugs together with low doses of 2-APB or CBD to determine whether nonlinear additivity between ligands is a more general feature in these proteins.

## Materials and methods

### Cell culture

Human embryonic kidney cells (HEK293) from ATCC (CRL-1573) were kept at 37°C in an atmosphere with 5% $CO_2$ and grown in Dulbecco's modified Eagle's medium (DMEM) with high glucose, pyruvate, L-glutamine, and phenol red, supplemented with 10% fetal bovine serum (vol/vol) and 10 mg/mL gentamicin. For transfection, cells were detached with trypsin, resuspended in DMEM, and seeded onto No. 1 glass coverslips in 3 mL dishes at 10–40% confluency. Transfections were performed on the same day using FuGENE6 Transfection Reagent (Roche Applied Science, Madison, WI). TRP channel constructs were co-transfected with pGreen-Lantern (Invitrogen, Carlsbad, CA) at a ratio of 2:1 to visualize successfully transfected cells. Electrophysiological recordings were done 18–36 hr after transfection. Cells tested negative for Mycoplasma infection, and were not authenticated; cell type is not critical for this study, as the most relevant aspect concerning our observations is that there are no contaminating currents that might me confounded for the TRPV1, TRPV2, or TRPV3 channels studied here. To address this, we performed all the necessary experiments to show that no background currents exist in our cell line that respond to any of the test compounds (CBD and 2-APB) used in this study.

### Molecular biology

The WT rat TRPV1 (*Caterina et al., 1997*) and TRPV2 (*Caterina et al., 1999*) channel cDNA were provided by Dr. David Julius (UCSF, CA), and mouse TRPV3 (*Peier et al., 2002*) was provided by Dr. Feng Qin (SUNY Buffalo, NY). All constructs were cloned into modified pcDNA3.1(+) and pcDNA1 for high and low levels of expression, respectively. CBD binding site mutants were generated using the two-step PCR method using Phusion High-Fidelity DNA polymerase (New England Biolabs, Ipswich, MA), T4 ligase Quick Ligation kit (New England Biolabs) and NovaBlue Singles competent cells (MilliporeSigma, Burlington, MA) and Sanger-sequenced to check for PCR errors. Cumulative pore mutants were generated by Gibson assembly (GeneArt Gibson Assembly kit, Thermo Fisher Scientific, Waltham, MA) following the manufacturer's instructions and using a gBlock (Integrated DNA Technologies, Coralville, IA) encompassing nucleotides 1695–1993 in the rTRPV1 coding region.

### Patch-clamp electrophysiology

Patch-clamp recordings were performed on transiently transfected HEK293 cells at room temperature (21–23°C) unless stated otherwise. Data were acquired with an Axopatch 200B amplifier (Molecular Devices, Sunnyvale, CA) and digitized with a Digidata1550B interface and pClamp10 software

(Molecular Devices). All data were analyzed using Igor Pro 8.04 (Wavemetrics, Portland, OR). Pipettes were pulled from borosilicate glass (1.5 mm O.D. × 0.86 mm I.D. × 75 mm L; Harvard Apparatus) using a Sutter P-97 puller and heat-polished to final resistances between 0.5 and 4 MΩ using a MF-200 microforge (World Precision Instruments). 90% series resistance ($R_s$) compensation was applied in all whole-cell recordings except those involving changes in temperature. An agar bridge (1 M KCl; 4% weight/vol agar; teflon tubing) was used to connect the ground electrode chamber and the main recording chamber.

The extracellular solution consisted of (mM) 130 NaCl, 10 HEPES, 10 EGTA, pH 7.4 (NaOH/HCl). The same solution was used as the intracellular recording solution except for the addition of 10 $MgCl_2$, which was included to block endogenous inward-rectifying and low-threshold temperature-sensitive currents that were observed in some recordings. Stocks of 2-APB (1 M; Sigma-Aldrich, St. Louis, MO), CBD (10 mM; Cayman Chemical, Ann Arbor, MI) were prepared using DMSO, and capsaicin stock solutions (100 mM; Sigma-Aldrich) were prepared in ethanol. 2-APB stocks were prepared fresh every day, and CBD stocks were aliquoted and stored at –20°C, and diluted in recording solution immediately prior to recordings. The effective concentration of 2-APB in some of the most concentrated recording solutions that we used might be overestimated because they were close to the solubility limit of the compound. Probenecid stock solution (100 mM) was made in 1 M NaOH, and used at a final concentration of 100 μM in recording solution. We verified the pH of solutions using pH-indicator paper.

Whole-cell and excised-patch data using gap-free recordings were acquired at 5 kHz and low-pass filtered at 1 kHz. Current–voltage (I–V) relations were acquired at 10 kHz and low-pass filtered at 2 kHz, and were obtained by applying 300-ms-long voltage steps from a holding potential of 0 mV with a frequency of 1 Hz. Test pulses went from –100 to +100 mV in 10 mV increments. In all experiments at room temperature, a gravity-fed perfusion system (RSC-200, BioLogic, France) was used, in which outlets of glass capillaries were placed right in front of the recorded cells, and a motorized holder was used to switch between tube outlets. For rTRPV2 channel concentration–response relations for CBD and 2-APB in the presence and absence of CBD (*Figure 1E and F*), we did not subtract background currents from the plotted data, which we normalized to the maximal concentration of either CBD (40 μM), 2-APB (4 mM), or 2-APB (4 mM) + 10 μM CBD. For the dose–response relations for WT and mutant rTRPV1 and rTRPV2 channels, as well as group data from gap-free recordings and measurements of sensitization, we subtracted the background currents measured in the absence of stimuli from the test currents before normalization. For normalization of the dose–response relations for 2-APB in some of the mutants, we used the concentration that yielded maximal currents because we observed pronounced channel desensitization at the highest concentrations in some of the mutants. Mean time courses of mTRPV3 channel activation were obtained by aligning currents from each experiment to the time in which cells were exposed to 2-APB for a second time (i.e. after the first exposure of 10 s duration that was used for normalization), segmented into intervals of 1.4 s duration (i.e. 7000 data points), normalized, and averaged. All group data are shown as mean ± SEM.

For single-channel recordings in the inside-out configuration, we acquired data at 10 kHz, low-pass filtered at 2 kHz, and we covered pipettes with dental wax to reduce capacitive transients and used pipettes with open tip resistances between 4 and 10 MΩ. Holding voltage was –80 mV, and 500 ms sweeps were recorded with a 100 ms inter-sweep interval, with 50 sweeps per patch in either control, 2-APB or CBD, and 2-APB and CBD applied together. All recorded traces were baseline-subtracted so that the mean current value in the absence of opening events was centered at 0 pA. The all-points histograms containing data from all sweeps per patch and experimental condition were normalized to the number of points at the peak centered at 0 pA and binned using 0.2 pA intervals.

Experiments involving temperature increases were carried out using a custom-built device as previously described (*Islas et al., 2015*; *Sánchez-Moreno et al., 2018*), which consists of using a wire-based microheater enclosed in a glass pipette for insulation and connected to a power source for heating. For our device, we used 0.1 mm diameter nichrome wire connected to a Keysight Technologies (Santa Rosa, CA) E3631A Triple Output DC Power Supply. Maximum output was utilized from the power supply, which was approximately 6 V and 0.74 A. The wire was introduced into a glass capillary pipette, which was bent to a U-shape using a flame. The U-shaped pipette with the wire inside connected to the power source was held on a micro-manipulator and placed right in front of the patch pipette inside the recording chamber. To estimate the changes in temperature achieved

by our heating device, we used the resistance measured from an open patch pipette, whose resistance measurements as a function of heating were previously calibrated by exchanging the recording bath with solutions at multiple temperatures, which we measured with a thermistor (Warner TC-324B, Hamden, CT) placed close to the pipette tip inside the bath. The heating device was then placed at a defined distance from the same pipette tip and the pipette resistance measured after turning the power supply on with maximal output for 90 s. For each calibration, we measured the pipette resistance $R$ as a function of bath temperature and fit the data to

$$R(\text{Temp}) = R_0 \times \exp(A_0/\text{Temp}) \tag{1}$$

where the fit parameters $R_0$ and $A_0$ are constants specific for each pipette. From *Equation 1*, we obtained the temperature (Temp) as a function of resistance:

$$\text{Temp} = A_0/[\ln(R) - \ln(R_0)] \tag{2}$$

We then introduced the heating device and recorded resistance as a function of time at full power source output for 90 s and fitted the resulting function to a double-exponential function of time:

$$R(\text{time}) = R(\text{time} = 0) + A_1 \times \exp(-\text{time} \times k_1) + A_2 \times \exp(-\text{time} \times k_2) \tag{3}$$

where the fitted parameters $A_1, A_2, k_1, K_2$, are constants specific to each heating device and its relative position to the tip of the pipette. The other fit parameter $R(\text{time} = 0)$ depends entirely on the pipette, and is equivalent to $R_0$ in *Equation 2* – the resistance of the pipette at the room temperature of 21°C. We obtained reproducible fitting parameters for the exponential using different pipettes with the same heating device, suggesting a highly reproducible time-dependent increase in temperature. To estimate the average temperature change during each experiment, we expressed *Equation 2* in terms of *Equation 3*:

$$\text{Temp(time)} = A_0[\ln(R(\text{time})) - \ln(R_0)] \tag{4}$$

and plotted our currents as a function of temperature by transforming recording time into temperature using *Equation 4* and the mean fit parameters for the exponential terms that we obtained from several trials with the same heating device and different open pipettes (n = 15). Based on this high reproducibility, we assume that the time dependence and magnitude of the temperature change was similar in all experiments, and thus used the same parameters for all experiments. $Q_{10}$ values were calculated by taking the logarithm of current–temperature relations from individual experiments, fitting them to a line with slope $m$ over the range from 30–40°C, and then using

$$Q_{10} = 10^{10 \times m}. \tag{5}$$

## High-performance liquid chromatography

Samples were prepared using our regular recordings solution without adding trifluoroacetic acid to maintain a neutral pH. We injected 50 nmol of 2-APB and 5 nmol of CBD, either by themselves or together, and the final volume injected into the HPLC column was 400 µL with the compound diluted in dH$_2$O. We injected samples into a 5 µM Ultrasphere C18 column (Beckman Coulter, Brea, CA) on a 1525 Binary pump HPLC system (Waters Corporation, Milford, MA), and used a gradient of 0% to 100% acetonitrile over 40 min and monitored sample absorbance at 228 nm.

## Sequence alignment and structural model depiction

We generated our amino acid sequence alignments using Jalview software (University of Dundee).

## rTPRV2 channel expression using Baculovirus and mammalian expression system

To produce the rTRPV2 channel for cryo-EM, the channel was cloned into the pEG vector in which EGFP was substituted with mVenus (*Rana et al., 2018*) and expressed in tsA201 cells using the previously published Baculovirus-mammalian expression system with a few minor modifications (*Goehring et al., 2014*). Briefly, P1 virus was generated by transfecting Sf9 cells (~2.5 million cells on a T25 flask with a vent cap) with 50–100 ng of fresh Bacmid using Cellfectin. After 4–5 d incubation in a

humidified 28°C incubator, the cell culture media was collected by centrifugation (3000 × $g$ × 10 min), supplemented with 2% FBS, and filtered through a 0.45 µm filter to harvest the P1 virus. To amplify the P1 virus, ~500 mL Sf9 cell cultures at an ~1.5 million cells/mL density were infected with 1–200 µL of the virus and incubated in a 28°C shaking incubator for 3 d. The cell culture media was then collected by centrifugation (5000 × $g$ 20 min), supplemented with 2% FBS, and filter through 0.45 µm filter to harvest P2 virus. The volume of P1 virus used for the amplification was determined by carrying out a small-scale amplification screening in which ~10 mL Sf9 cell cultures at the same density were infected with different volume of P1 virus and harvested after 3 d to transduce tsA201 cells and compare the expression level of rTRPV2 using mVenus fluorescence intensity. The P2 virus was protected from light using aluminum foil and stored at 4°C until use. To express the rTRPV2, tsA201 cells at ~1.5 million cells/mL in Freestyle medium with 2% FBS were transduced with 10% (v/v) P2 virus and incubated at a 37°C CO₂ incubator. To boost the protein expression, sodium butyrate (2 M stock in H₂O) was added to 10 mM at ~16 hr of post-transduction. The culture was continued at the 37°C CO₂ incubator for another 24 hr, and the cells were harvested by centrifugation (5000 × $g$ × 20 min) and frozen at –80°C until use.

## rTRPV2 channel purification

Prior to extraction of rTRPV2 from tsA201 cells, membrane fractionation was carried out using a hypotonic solution and ultracentrifugation. The cells were first resuspended in a hypotonic solution (20 mM Tris pH 7.5 and 10 mM NaCl) with protease inhibitors (pepstatin, aprotinin, leupeptin, benzamidine, trypsin inhibitor, PMFS) using a Dounce homogenizer, incubated at 4°C for ~30 min, and centrifuged at 1000 × $g$ for 10 min to remove cell debris. The supernatant was ultracentrifuged for 1 hr (45,000 rpm, Beckman Ti45 rotor) and collected membranes were stored at –80°C until use. To purify rTRPV2, the fractionated membranes were resuspended in an extraction buffer (50 mM Tris pH 7.5, 150 mM NaCl, 5% glycerol, 2 mM TCEP, 50 mM DDM, 5 mM CHS with the protease inhibitor mixture used above) and extracted for 1 hr at 4°C. The solution was clarified by centrifugation (12,000 × $g$ × 10 min) and incubated with CoTALON resins at 4°C for 1 hr, at which point the mixture was transferred to an empty disposable column (Econo-Pac Bio-Rad). The resin was washed with 10 column volume of Buffer A (50 mM Tris pH 7.5, 150 mM NaCl, 1 mM DDM, 0.1 mM CHS, and 0.1 mg/mL porcine brain total lipid extract) with 10 mM imidazole, and bound proteins were eluted with Buffer A with 250 mM imidazole. The eluate was concentrated using Amicon Ultra (100 kDa) to ~350–450 µL and loaded onto a Superose6 (10 × 300 mm) gel filtration column and separated with Buffer A. All purification steps described above was carried out at 4°C or on ice.

## Lipid nanodisc reconstitution of the rTRPV2 channel

Lipid nanodisc reconstitution was performed following the previously published methods with minor modifications (*Matthies et al., 2018*; *Tan et al., 2022*). On the day of nanodisc reconstitution, the rTRPV2 channel purified by Superose6 in detergent was concentrated to ~1–3 mg/mL and incubated with histidine-tagged MSP1E3D1 and 3:1:1 mixture of 1-palmitoyl-2-oleoyl-sn-glycero-3-phosphocholine (POPC), 1-palmitoyl-2-oleoyl-sn-glycero-3-phospho-(1'-rac-glycerol) (POPG), and 1-palmitoyl-2-oleoyl-sn-glycero-3-phosphoethanolamine (POPE) for 30 min at room temperature. The mixture was transferred to a tube with Biobeads (~30–50-fold of detergent; w/w) and incubated at room temperature for ~3 hr in the presence of TEV protease (prepared in-house) and 2 mM TCEP to remove N-terminal fusion protein including poly-histidine and mVenus tag. The reconstituted protein was loaded onto Superose6 column (10 × 300 mm) and separated using 20 mM Tris and 150 mM NaCl buffer at 4°C. The success of nanodisc reconstitution was confirmed by collecting separated fractions and running SDS-PAGE to verify the presence of rTRPV2 and MSP1E3D1 bands at a similar ratio. Typically, optimal reconstitution required the incubation of 1:10:200 or 1:10:400 molar ratio of tetrameric rTRPV2, MSP1E3D1, and the lipid mixture.

## Cryo-EM sample preparation and data acquisition

6.5 mg/mL TRPV2 in nanodiscs were incubated with 100 µM CBD and 1 mM 2-APB on ice for 30 min and then 3 µL aliquots were applied to glow-discharged Quantifoil grids (R 1.2/1.3 Cu 300 mesh). The grids were blotted for 4 s, with blot force of 4 and 100% humidity, at 16°C using an FEI Vitrobot Mark IV (Thermo Fisher Scientific), followed by plunging into liquid ethane cooled by liquid nitrogen.

Images were acquired using an FEI Titan Krios equipped with a Gatan LS image energy filter (slit width, 20 eV) operating at 300 kV. A Gatan K3 Summit direct electron detector was used to record movies in superresolution mode with a nominal magnification of ×105,000,, resulting in a calibrated pixel size of 0.415 Å per pixel. The typical defocus values ranged from −0.5 to −1.5 μm. Exposures of 1.6 s were dose-fractionated into 32 frames, resulting in a total dose of 52 $e^-$ Å$^{-2}$. Images were recorded using the automated acquisition program SerialEM (*Mastronarde, 2005*).

## Image processing

All processing was completed in RELION (*Mastronarde, 2005*). The beam-induced image motion between frames of each dose-fractionated micrograph was corrected and binned by 2 using MotionCor2 (*Zheng et al., 2017*) and contrast transfer function (CTF) estimation was performed using CTFFIND4 (*Rohou and Grigorieff, 2015*). Micrographs were selected, and those with outliers in defocus value and astigmatism, as well as low resolution (>5 Å) reported by CTFFIND4 were removed. The initial set of particles from 300 micrographs were picked using Gautomatch (www2.mrc-lmb. cam.ac.uk/research/locally-developed-software/zhang-software/#gauto) and followed by reference-free two-dimensional (2D) classification in RELION. The good classes were then used as template to pick particles from all selected micrographs using Gautomatch. Particles (1,665,271) were picked and extracted with 2× downscaling (pixel size, 1.66 Å). Several rounds of reference-free 2D classification were performed to remove ice spot, contaminants, and bad particles. The good particles were 3D classified with C4 symmetry using reference generated by 3D initial model. Good class (380,829) were then selected and reextracted without binning (pixel size, 0.83 Å) followed by 3D auto-refine. After that, the refined particles were expanded from C4 symmetry to C1 symmetry and then subjected to 3D Classification (skip alignment) with transmembrane domain mask. Six classes show one CBD binding site in each monomer (conformation A) and one class shows two CBD binding sites per monomer (conformation B). Particles (1,370,377) from these six classes were combined and duplication was removed. Finally, 321,717 unique particles were obtained and submitted to final step of 3D auto-refine with C4 symmetry. Particles (43,071) from conformation B were selected by removing duplication followed by 3D auto-refine. The final reconstruction was reported at 3.23 Å for conformation A and 3.32 Å for conformation B.

## Model building and structure refinement

Model building was first carried out by manually fitting the monomer of rTRPV2 (PDB 6U84) into the electron microscopy density map using UCSF Chimera (*Pettersen et al., 2004*). The model was then manually built in Coot (*Emsley et al., 2010*) and refined using real space refinement in PHENIX (*Adams et al., 2010*) with secondary structure and geometry restraints. The final model was evaluated by comprehensive validation in PHENIX. Structural figures were generated using PyMOL (https:// pymol.org/2/support.html) and UCSF Chimera.

## Acknowledgements

We thank Rick Aldrich, Eric Senning, and Marcel Goldschen-Ohm for helpful discussions, León D Islas and Ernesto Ladrón-de-Guevara for discussions in building the heating device, and Huaibin Wang in the NIH Multi-Institute Cryo-EM Facility (MICEF) for assistance in acquiring cryo-EM data. This work utilized NIDDK Cryo-EM Core Facility, the NIH MICEF, and computational resources of the NIH HPC Biowulf cluster (http://hpc.nih.gov). This research was supported by startup funds from the University of Texas at Austin (to AJO), NINDS R00 Career Development Award 4R00NS101053-02 (to AJO), and the Intramural Research Programs of the NINDS (to KJS).

## Additional information

### Competing interests

Chanhyung Bae: Currently affiliated with Janssen R&D. The author has no financial interests to declare. Kenton J Swartz: Senior editor, *eLife*. Andres Jara-Oseguera: Reviewing editor, *eLife*. The other authors declare that no competing interests exist.

## Funding

| Funder | Grant reference number | Author |
| --- | --- | --- |
| University of Texas at Austin | Startup funds | Andres Jara-Oseguera |
| National Institute of Neurological Disorders and Stroke | 4R00NS101053-02 | Andres Jara-Oseguera |
| National Institute of Neurological Disorders and Stroke | Intramural Research Programs | Kenton J Swartz |

The funders had no role in study design, data collection and interpretation, or the decision to submit the work for publication.

## Author contributions

Aaron Gochman, Conceptualization, Data curation, Formal analysis, Investigation, Visualization, Methodology, Writing – original draft, Writing – review and editing; Xiao-Feng Tan, Data curation, Formal analysis, Validation, Investigation, Visualization, Methodology, Writing – original draft, Writing – review and editing; Chanhyung Bae, Helen Chen, Investigation, Methodology; Kenton J Swartz, Supervision, Funding acquisition, Project administration, Writing – review and editing; Andres Jara-Oseguera, Conceptualization, Resources, Data curation, Software, Formal analysis, Supervision, Funding acquisition, Validation, Investigation, Visualization, Methodology, Writing – original draft, Project administration, Writing – review and editing

## Author ORCIDs

Aaron Gochman  http://orcid.org/0000-0003-2672-3226
Xiao-Feng Tan  http://orcid.org/0000-0001-9327-2424
Kenton J Swartz  http://orcid.org/0000-0003-3419-0765
Andres Jara-Oseguera  http://orcid.org/0000-0001-5921-9320

## Decision letter and Author response

Decision letter https://doi.org/10.7554/eLife.86166.sa1
Author response https://doi.org/10.7554/eLife.86166.sa2

# Additional files

## Supplementary files

• MDAR checklist

## Data availability

All data needed to evaluate the conclusions in the article are present in the article, the supplementary materials, or uploaded as source data. Maps for conformations A and B for rTRPV2 have been deposited in the Electron Microscopy Data Bank (EMDB) under accession codes EMD-40582 and EMD-40583, respectively. Models of conformations A and B for rTRPV2 have been deposited in the Protein Data Bank with accession codes 8SLX and 8SLY, respectively.

The following datasets were generated:

| Author(s) | Year | Dataset title | Dataset URL | Database and Identifier |
| --- | --- | --- | --- | --- |
| Swartz KJ, Tan X | 2023 | Rat TRPV2 bound with 1 CBD ligand in nanodiscs | https://www.ebi.ac.uk/emdb/EMD-40582 | Electron Microscopy Data Bank, EMD-40582 |
| Swartz KJ, Tan X | 2023 | Rat TRPV2 bound with 2 CBD ligands in nanodiscs | https://www.ebi.ac.uk/emdb/EMD-40583 | Electron Microscopy Data Bank, EMD-40583 |
| Tan X, Swartz KJ | 2023 | Rat TRPV2 bound with 1 CBD ligand in nanodiscs | https://www.rcsb.org/structure/8SLX | RCSB Protein Data Bank, 8SLX |
| Tan X, Swartz KJ | 2023 | Rat TRPV2 bound with 2 CBD ligands in nanodiscs | https://www.rcsb.org/structure/8SLY | RCSB Protein Data Bank, 8SLY |

The following previously published datasets were used:

| Author(s) | Year | Dataset title | Dataset URL | Database and Identifier |
|---|---|---|---|---|
| Pumroy RA, Moiseenkova-Bell VY | 2019 | Apo full-length rat TRPV2 in nanodiscs, state 1 | https://www.rcsb.org/structure/6U84 | RCSB Protein Data Bank, 6U84 |
| Pumroy RA, Protopopova AD, Gallo PN, Moiseenkova-Bell VY | 2022 | Activated state of 2-APB and CBD-bound wildtype rat TRPV2 in nanodiscs | https://www.rcsb.org/structure/7T37 | RCSB Protein Data Bank, 7T37 |
| Nan NS, Fan Y | 2022 | Cholesterol bound state of mTRPV2 | https://www.rcsb.org/structure/7XEM | RCSB Protein Data Bank, 7XEM |
| Zhang K, Julius D, Cheng Y | 2021 | cryo-EM structure of unliganded minimal TRPV1 | https://www.rcsb.org/structure/7L2P | RCSB Protein Data Bank, 7L2P |
| Ruth A P, Amrita S, Liu Y, Taylor ETH, Zhao S, Yevgen Y, Tibor R, Han S, Vera YMB | 2019 | CBD-bound full-length rat TRPV2 in nanodiscs, state 1 | https://www.rcsb.org/structure/6U8A | RCSB Protein Data Bank, 6U8A |
| Pumroy RA, Protopopova AD, Gallo PN, Moiseenkova-Bell VY | 2022 | Inactivated state of 2-APB-bound wildtype rat TRPV2 in nanodiscs | https://www.rcsb.org/structure/7N0M | RCSB Protein Data Bank, 7N0M |
| Gao Y, Cao E, Julius D, Cheng Y | 2016 | Structure of TRPV1 determined in lipid nanodisc | https://www.rcsb.org/structure/5IRZ | RCSB Protein Data Bank, 5IRZ |

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
