## [Editor Report]

This is an important report on the discovery of a strong sensitizing effect of cannabidiol on the activation of TRPV2 channels by 2-APB. The conclusions are convincingly supported by electrophysiological recordings and cryo-EM structures, but identification of a clear molecular mechanism will require additional structural work. The paper will be of interest to the ion channel research community.

---

## [Decision Letter]

**Decision letter after peer review:**

Thank you for submitting your article "Cannabidiol sensitizes TRPV2 channels to activation by 2-APB" for consideration by *eLife*. Your article has been reviewed by 3 peer reviewers, including László Csanády as the Reviewing Editor and Reviewer #1, and the evaluation has been overseen by Volker Dötsch as the Senior Editor. The following individual involved in the review of your submission has agreed to reveal their identity: Wei Lü (Reviewer #2).

Essential revisions:

1. The authors obtained ligand densities at positions different from those observed in previous TRPV2 studies. Therefore, given that their structures were obtained under different experimental conditions, it seems inappropriate to use previously published apo structures as a control. It is therefore essential to obtain an apo rTRPV2 structure as a control, to support the assignment of the observed ligand densities.

2. The authors have identified two distinct CBD binding sites, but the manuscript does not discuss the functional roles of those sites. Two of the positions they have targeted so far (L538, L541) seem to be part of the single CBD site observed in conformation A, whereas the third position (M532) is near the internal CBD site but the side chain is pointing in the opposite direction. A more systematic screening of the two CBD binding sites through electrophysiological assessment of mutants seems essential to evaluate the roles of both sites, specifically to investigate whether both sites are necessary for the observed sensitization of TRPV2 by CBD.

3. There is a strong density within the S1-S4 domain. That density was not modeled in the structure and is not mentioned in the manuscript. Is it possible that this density represents a bound 2-APB molecule? Although a complete 2-APB molecule cannot fit into the density, a portion of 2-APB could reasonably fit, leaving for example one of the phenyl groups outside of the density, potentially due to its high flexibility. Such a possibility would be further supported if the density didn't show up in the apo structure (see 1. above). In that case, studying the residues around it by targeted mutagenesis and patch clamp electrophysiology would be recommended.

*Reviewer #1 (Recommendations for the authors):*

I have only a few formal comments regarding data display that the authors may consider.

Figures 1G (bottom right), 1H, 4F, 4K (right), Figure 7C, Figure 6 – Figure Suppl. 1F-G:

Bar charts with logarithmic ordinates should perhaps be replaced by dot plots (the bar representation is visually misleading, as the plotted bar height is arbitrary…)

Figure 2: The heat-map representations of Po and amplitude histograms are unusual. For me, it is much harder to obtain a quantitative impression of the effects from these plots compared to conventional histogram representations.

Figure 4: In all panels please indicate the channel construct used (rTRPV1 or mTRPV3)

Figure 4J: Please indicate voltage also for the second part of the experiment.

Figure 4K left: Please indicate in the legend what the magenta symbols represent. (The legend just states: "data" were fit to a mono-exponential function of time)

Figure 6G: This 2-D graph with two different axes for each dimension and two types of symbols for each of these axes is impenetrable. Some sort of simpler representation, perhaps even breaking the graph up into multiple panels if necessary, would make it easier for the reader to grab the message.

Figure 3 – Figure Suppl. 1, end of legend: rTRPV2, not rTRPV1

*Reviewer #2 (Recommendations for the authors):*

Here are some comments that the authors may wish to consider.

1. There is a strong density within the S1-S4 domain. This density was not modeled in the structure and is not mentioned in the manuscript. Is it possible that this density represents a bound 2-APB molecule? Although a complete 2-APB molecule cannot fit into the density, a portion of 2-APB could reasonably fit, leaving for example one of the phenyl groups outside of the density, potentially due to its high flexibility. It is worth noting that the hydrophobic tails of the two CBD molecules in this study also lack densities, which should be acknowledged in the manuscript. While I highly recommend further investigation of the nature of this density by studying the residues around it using patch clamp electrophysiology, it's up to the authors to decide whether they want to do this.

2. The authors generated mutants in residues that are close to the CBD binding sites to investigate the mechanism of CBD-dependent sensitization. It should be noted that V352, one of the mutated residues, is not in direct contact with CBD, and is actually located on the opposite side of the S5 helix from the CBD binding site. On the other hand, V352 faces (but does not form direct contact with) the proposed 2-APB binding site by Pumroy et al. 2022, although a 2-APB molecule is not unambiguously supported by the density in that study, as noted by the authors of this manuscript. Therefore, caution should be taken when interpreting the electrophysiology data of the V532 mutant.

3. It is interesting to note that the authors have identified two distinct CBD binding sites. However, the manuscript does not discuss the functional roles of these sites. It would be informative to investigate whether both sites are necessary for the observed sensitization of TRPV2 by CBD, which would provide a more detailed understanding of the mechanism by which CBD sensitizes TRPV2.

4. Although the nominal resolution of conformation A is higher than that of conformation B, it is clear that the map quality of B is substantially higher than that of A. This should be mentioned in the manuscript.

5. The authors discussed three possibilities as to why the extracellular CBD site is present in more protein particles than the intracellular CBD site. However, it is also possible (perhaps more likely) that the difference is caused by delipidation since an important part of the intracellular CBD binding site is formed by a lipid molecule that was modeled in the structure. The identity of this lipid is unclear and may be missing in the lipid mixture used for nanodisc reconstitution. In fact, the authors assigned the lipid as phosphatidylinositol, which was not added in their nanodisc reconstitution.

6. The description of how conformations A and B were obtained reads a bit confusing. The text suggests B was obtained by further classification of particles used to refine A. But Figure 3 – —figure supplement 2 implies that both conformations were obtained in the same TMD-focused classification. Furthermore, it is unclear why the first three gray classes in the TMD-focused classification were shown twice in Figure 3 – —figure supplement 2.

7. The visual representation of Figure 4K and 6G can be improved for better readability. Similarly, in Figure 3B, the two CBD molecules and the lipid molecule should be labeled for easy identification.

*Reviewer #3 (Recommendations for the authors):*

Essential additions for presented experiments:

• Obtain control structures of TRPV2 under the following conditions: apo, in the presence of only CBD, in the presence of only 2-APB. As the authors got a different result in their structure with 2-APB and CBD from previous work, they must do controls to show how their preparation of TRPV2 behaves as a baseline and how each individual drug impacts the channel. They report no density for 2-APB in their structures, but it remains possible that 2-APB on its own may cause conformation changes. Additionally, these controls would show whether 2-APB is necessary for the second 2-APB site to be accessible.

• As one of the TRPV2 mutation sites (V532M) showed no response to either 2-APB or CBD, the authors should do controls to determine via other stimuli whether the loss of response is specific to these drugs or due to a dead channel.

Technical/Presentation Issues:

• As a technical note for the structural work, the pixel size of the cryo-EM maps is incorrect. When compared to the 2019 and 2022 maps of TRPV2 with CBD and maps of apo TRPV1, the maps from this paper were noticeably larger. A crude scaling based on a structure of the TRPV2 ankyrin repeat domains obtained by x-ray crystallography suggests that the correct pixel size is around 0.83 Å/pix rather than 0.86 Å/pix, a significant difference in scale that needs to be corrected.

• The authors have assigned the extra densities in the vanilloid pocket in their conformation B as a π lipid and CBD. They should provide figures showing the fit of these molecules as well as other possibilities, specifically other lipid headgroups such as PC for the lipid density and 2-APB for the CBD density. Given the conical rather than flat shape of the headgroup density and the incorrect scale of the map, it seems unlikely that the lipid in conformation B is PI.

Suggestions:

• This paper seems to be pulling in two directions at once. On one hand, the stated goal of the paper is an investigation into the mechanisms of TRPV2 sensitization by CBD, on the other hand, the authors seem primarily interested in investigating it only from the perspective of how the group of TRPV1-TRPV3 responds to the same form of sensitization. I think the authors should pick one direction and stick with it. If they want to focus on TRPV2, they could examine TRPV2 orthologs and/or examine how CBD sensitization interacts with the other reported forms of sensitization. Alternatively, they could do a more in-depth comparative examination of the response of TRPV1-3 to CBD and 2-APB, ideally including more structural work. With either of these approaches, even if the authors are ultimately unable to track down the exact molecular basis of sensitization, they would at least have produced data that can be meaningfully interpreted for future studies on the topic.

• Throughout the entire paper, the authors use a single orthologue of three different channels, rat for TRPV1 and TRPV2, and mouse for TRPV3. If the author's interest is in tracking down the mechanisms of channel sensitization, it might be a more productive approach to examine several orthologues of the same channel to see if there are varying degrees of sensitization by CBD, a feasible possibility as TRPV2 orthologs are known to respond to 2-APB differently. That would significantly reduce variables when introducing mutations to test the effect on channel sensitization. An approach like this was successfully used to examine heat desensitization of TRPV1 (Luo, Wang, Li, et al., Nat Commun 2019).

• Zubcevic et al., *eLife* 2019 did a very nice job of tracing channel hysteresis to the conformation of the C-terminal loop and showed that the channel could be rendered completely sensitized to 2-APB by disrupting a single salt bridge with either K169A or E751A. This was tied to structural work showing a change in channel conformation, and the presence of density for 2-APB, in a structure with that mutation. This work is directly relevant to the observations in this paper, particularly as TRPV2 conserves those two residues and the conformational range at the C-terminus as observed in TRPV3. The authors may wish to investigate how this mutation interacts with 2-APB and CBD sensitization – is TRPV2 sensitized to the same extent as TRPV3 with this mutation? Does CBD still further sensitize either TRPV2 or TRPV3 with this mutation?

• Work in Mo et al., *eLife* 2022 showed that Tyr phosphorylation can also sensitize TRPV2. The authors should mention this in their paper and may wish to investigate how this form of sensitization, as well as the methionine oxidation sensitization they already mention in the text, interacts with CBD sensitization.

---

## [Author Response]

Essential revisions:1. The authors obtained ligand densities at positions different from those observed in previous TRPV2 studies. Therefore, given that their structures were obtained under different experimental conditions, it seems inappropriate to use previously published apo structures as a control. It is therefore essential to obtain an apo rTRPV2 structure as a control, to support the assignment of the observed ligand densities.

We respectfully disagree. The only assignments of ligand densities we report here are for CBD. Under our experimental conditions, the majority of picked particles represent conformation A, which is practically identical (RSMD 0.647 Å) to the previous structure of rTRPV2 bound to CBD (PDB:6U88) and shows an equivalent non-protein density within the CBD site. A smaller fraction of our particles represents a newly observed conformation (conformation B), which nonetheless also has a non-protein density within the previously identified CBD binding site that is fully consistent with the assigned ligand. Together, our observations provide strong support for the previously proposed location of the CBD binding site, which was one of our stated goals for obtaining structural information.

Our map of conformation B contains an additional non-protein density that we assigned to CBD. We have now expanded Figure 3 to more clearly show the two densities we assigned to CBD and show how the density is better fit by CBD than 2-APB. As pointed out by reviewer 2, the two densities we assign to CBD in conformation B are higher quality compared to conformation A (and when compared to previously published CBD densities). In the revised manuscript we now more clearly articulate why we think the new density corresponds to CBD.

We also observed potentially interesting non-protein densities at other sites in the protein, which we did not assign, including a density within the S1-S4 domain. We agree that having our own apo structures might have allowed us to assign some of these densities. However, we carefully inspected all the previously published maps and found that contrary to the CBD-assigned density in conformation B, which is unique to our map, the rest of the unassigned densities can also be observed in several published TRPV2 structures, including an apo map (PDB:6U84, EMD-20677), a CBD-bound map (PDB:6U8A, EMD-20686) and a 2-APB bound map (PDB:7N0M, EMD-24109), suggesting that they do not represent CBD or 2-APB. We have modified Figure 3 – Figure Supp 5 to add a direct comparison with previous structures for the extra density within the S1-S4 domain, which are indistinguishable from ours.

2. The authors have identified two distinct CBD binding sites, but the manuscript does not discuss the functional roles of those sites. Two of the positions they have targeted so far (L538, L541) seem to be part of the single CBD site observed in conformation A, whereas the third position (M532) is near the internal CBD site but the side chain is pointing in the opposite direction. A more systematic screening of the two CBD binding sites through electrophysiological assessment of mutants seems essential to evaluate the roles of both sites, specifically to investigate whether both sites are necessary for the observed sensitization of TRPV2 by CBD.

We agree that it would in principle have been interesting to carefully interrogate the new CBD site using mutagenesis and patch-clamp recording. However, our interrogation of the previously published site did not help to localize determinants that explain the differential sensitivity between TRPV1 and TRPV2 channels to CBD, diminishing our enthusiasm for further interrogation of the new site.

We found that mutations at position V532 (in TRPV2) and M572 (in TRPV1) had the most marked effects of all mutations on the observed currents activated by 2-APB, completely ablating currents in the case of TRPV2 and dramatically reducing both the apparent affinity for 2-APB and the sensitization caused by CBD in the case of TRPV1. There is no structural evidence suggesting that 2-APB or CBD interact directly with the side-chains at V532 or M572, so the effect of mutations at that those sites must be allosteric. Even if mutations introduced at the second tentative CBD site were shown to affect CBD-dependent sensitization of 2-APB responses in TRPV2 channels, it would be impossible to determine whether the observed effect of the mutation is due to an allosteric effect or a direct disruption of the interaction between the channel and the CBD molecule. Further, it seems unlikely that structural determinants in that region are responsible for the large sensitizing effect of CBD on rTRPV2 vs the small effect on rTRPV1, since mutations at all non-conserved residues (including E570Q near the second CBD site) failed to confer a gain-of- function phenotype to TRPV1 (see Figure 7B).

We tested the strength of CBD-dependent sensitization in two TRPV2 mutants containing single residue substitutions at positions near the second CBD site, rTRPV2+Q530A and rTRV2+Q530G (new Figure 6—figure supplementFigure Suppl. 3), and found them to be indistinguishable from WT. We have modified the text to make it clearer that the functional relevance for the second CBD site remains unknown, but we still consider it noteworthy as an observation because of the possibility that other molecules, including endogenous ligands, could regulate channel function by interacting with that site. We also agree that thorough exploration of the role of the S4-S5 linker region on 2-APB and CBD sensitivity in TRPV1 and TRPV2 channels is warranted, but we consider this to be beyond the scope of this work. We think this body of work still makes an important contribution to the field and as both reviewer 1 and 2 state, the data presented supports the key conclusions we have formulated.

3. There is a strong density within the S1-S4 domain. That density was not modeled in the structure and is not mentioned in the manuscript. Is it possible that this density represents a bound 2-APB molecule? Although a complete 2-APB molecule cannot fit into the density, a portion of 2-APB could reasonably fit, leaving for example one of the phenyl groups outside of the density, potentially due to its high flexibility. Such a possibility would be further supported if the density didn't show up in the apo structure (see 1. above). In that case, studying the residues around it by targeted mutagenesis and patch clamp electrophysiology would be recommended.

We have inspected previously published maps of apo TRPV2 (PDB:6U84, EMD-20677), CBD-bound TRPV2 (PDB:6U8A, EMD-20686) and 2-APB bound TRPV2 (PDB:7N0M, EMD-24109) and this density within the S1-S4 domain is clearly seen in each. We originally commented on this in the legend to Figure 3-Figure Supp. 5, but we have revised that figure to clearly show how similar the density is between our structure and that of apo TRPV2 and we have added a brief discussion to the Results section.

Reviewer #1 (Recommendations for the authors):I have only a few formal comments regarding data display that the authors may consider.Figures 1G (bottom right), 1H, 4F, 4K (right), Figure 7C, Figure 6 – Figure Suppl. 1F-G:Bar charts with logarithmic ordinates should perhaps be replaced by dot plots (the bar representation is visually misleading, as the plotted bar height is arbitrary…)

We thank the reviewer for this suggestion, and we have replaced all bar graphs by dot plots. We have also included a visual reference denoting the value used for normalization, to better highlight the fold- increase in current activation by 2-APB caused by the addition of CBD.

Figure 2: The heat-map representations of Po and amplitude histograms are unusual. For me, it is much harder to obtain a quantitative impression of the effects from these plots compared to conventional histogram representations.

We thank the reviewer for this view, and we agree that the way the data are presented is not ideal for a quantitative assessment of the results. However, we consider that our choice of data visualization is well suited for qualitative comparisons between key experimental conditions using out entire dataset. We think this is appropriate because most recordings under all conditions except in the presence of 2- APB+CBD lack any channel openings, precluding quantitation of the NPo. We believe that the heat maps most clearly show that channel openings are almost exclusively observed when CBD and 2-APB are applied together in rTRPV2-transfected cells.

Figure 4: In all panels please indicate the channel construct used (rTRPV1 or mTRPV3)Figure 4J: Please indicate voltage also for the second part of the experiment.

Done.

Figure 4K left: Please indicate in the legend what the magenta symbols represent. (The legend just states: "data" were fit to a mono-exponential function of time)

We thank the reviewer for drawing attention to this point that was not clearly explained in the previous version of the manuscript, which we have now corrected.

In experiments measuring the time-course of activation by 2-APB alone (as in Figure 4H), we repeatedly exposed cells to the agonist, and fit each of the resulting current responses to a bi-exponential function of time (red curves on Figure 4H). The slow and fast time-constants obtained from these fits are shown on Figure 4K in yellow and magenta, respectively. In the first stimulation both the fast and slow time- constants take similar values, indicating an absence of sensitized channels because cells have not yet been exposed to agonist. After a single stimulation with 2-APB, the fast time-constant adopts values similar to those obtained from mono-exponential fits to currents elicited by 2-APB in combination with CBD – this fast time-constant likely originates from the fraction of channels in the cell that have become sensitized by 2-APB. We think that the slow time constant reflects the slow process of sensitization by 2-APB. Consistently, the slow time-constants obtained from fits to individual stimulations with 2-APB are similar to the values obtained from mono-exponential fits to the time course spanning the entire experiment. Together these data indicate that channels activate fast when sensitized by either 2-APB or 2-APB + CBD, but the latter experimental condition allows channels to reach the sensitized state much more rapidly.

Figure 6G: This 2-D graph with two different axes for each dimension and two types of symbols for each of these axes is impenetrable. Some sort of simpler representation, perhaps even breaking the graph up into multiple panels if necessary, would make it easier for the reader to grab the message.

We thank the reviewer for pointing this out. In the revised version, Figure 6 has been simplified and we have included an additional Figure Supplement (Figure 6—figure supplementFigure Supplement 2), allowing us to break down the complicated graphs into multiple, simpler graphs.

Figure 3 – Figure Suppl. 1, end of legend: rTRPV2, not rTRPV1

Thanks for catching this typo.

Reviewer #2 (Recommendations for the authors):Here are some comments that the authors may wish to consider.1. There is a strong density within the S1-S4 domain. This density was not modeled in the structure and is not mentioned in the manuscript. Is it possible that this density represents a bound 2-APB molecule? Although a complete 2-APB molecule cannot fit into the density, a portion of 2-APB could reasonably fit, leaving for example one of the phenyl groups outside of the density, potentially due to its high flexibility. It is worth noting that the hydrophobic tails of the two CBD molecules in this study also lack densities, which should be acknowledged in the manuscript. While I highly recommend further investigation of the nature of this density by studying the residues around it using patch clamp electrophysiology, it's up to the authors to decide whether they want to do this.

We have inspected previously published maps of apo TRPV2 (PDB:6U84, EMD-20677), CBD-bound TRPV2 (PDB:6U8A, EMD-20686) and 2-APB bound TRPV2 (PDB:7N0M, EMD-24109) and this density within the S1-S4 domain is clearly seen there, so we are confident that it is not 2-APB. We originally commented on this in the legend to Figure 3–Figure Supp. 5, but we have revised that figure to clearly show how similar the density is between our structure and that of apo TRPV2 and we have added a brief discussion to the Results section.

2. The authors generated mutants in residues that are close to the CBD binding sites to investigate the mechanism of CBD-dependent sensitization. It should be noted that V352, one of the mutated residues, is not in direct contact with CBD, and is actually located on the opposite side of the S5 helix from the CBD binding site. On the other hand, V352 faces (but does not form direct contact with) the proposed 2-APB binding site by Pumroy et al. 2022, although a 2-APB molecule is not unambiguously supported by the density in that study, as noted by the authors of this manuscript. Therefore, caution should be taken when interpreting the electrophysiology data of the V532 mutant.

We have revised the text to be clearer about the position of V532 and have tempered any conclusions.

3. It is interesting to note that the authors have identified two distinct CBD binding sites. However, the manuscript does not discuss the functional roles of these sites. It would be informative to investigate whether both sites are necessary for the observed sensitization of TRPV2 by CBD, which would provide a more detailed understanding of the mechanism by which CBD sensitizes TRPV2.

We agree that it would in principle have been interesting to carefully interrogate the new CBD site using mutagenesis and functional approaches. However, our interrogation of the previously published CBD site did not help to localize determinants that explain the differential sensitivity of TRPV channels to CBD, diminishing our enthusiasm for further interrogation of the new site. We consider that the strong perturbations to the responses to 2-APB caused by substitutions at positions V532 or M572, whose side- chains are not expected to contact either CBD or 2-APB, are indicative that the region has strong allosteric control over channel responses to 2-APB. In the face of this, it would be challenging to provide singular mechanistic interpretations to experimental observations regarding the effects of mutants in this region. We agree that an exhaustive exploration of the role of the S4-S5 linker region in controlling activation by 2-APB and CBD in TRPV1 and TRPV2 channels is warranted, but we think this is beyond the scope of this manuscript.

4. Although the nominal resolution of conformation A is higher than that of conformation B, it is clear that the map quality of B is substantially higher than that of A. This should be mentioned in the manuscript.

We now comment on this in the Results section.

5. The authors discussed three possibilities as to why the extracellular CBD site is present in more protein particles than the intracellular CBD site. However, it is also possible (perhaps more likely) that the difference is caused by delipidation since an important part of the intracellular CBD binding site is formed by a lipid molecule that was modeled in the structure. The identity of this lipid is unclear and may be missing in the lipid mixture used for nanodisc reconstitution. In fact, the authors assigned the lipid as phosphatidylinositol, which was not added in their nanodisc reconstitution.

The reviewer raises a very interesting point, and we now discuss this in the Results section. Thank you!

6. The description of how conformations A and B were obtained reads a bit confusing. The text suggests B was obtained by further classification of particles used to refine A. But Figure 3 – —figure supplement 2 implies that both conformations were obtained in the same TMD-focused classification. Furthermore, it is unclear why the first three gray classes in the TMD-focused classification were shown twice in Figure 3 – —figure supplement 2.

We apologize for the confusion and we have now revised the text to be clear about how the two conformations were obtained.

7. The visual representation of Figure 4K and 6G can be improved for better readability. Similarly, in Figure 3B, the two CBD molecules and the lipid molecule should be labeled for easy identification.

We have made these changes to improve these three figures.

Reviewer #3 (Recommendations for the authors):Essential additions for presented experiments:• Obtain control structures of TRPV2 under the following conditions: apo, in the presence of only CBD, in the presence of only 2-APB. As the authors got a different result in their structure with 2-APB and CBD from previous work, they must do controls to show how their preparation of TRPV2 behaves as a baseline and how each individual drug impacts the channel. They report no density for 2-APB in their structures, but it remains possible that 2-APB on its own may cause conformation changes. Additionally, these controls would show whether 2-APB is necessary for the second 2-APB site to be accessible.

We respectfully disagree. The only assignments of ligand densities we report here are for CBD. Under our experimental conditions, the majority of picked particles represent conformation A, which is practically identical (RSMD 0.647 Å) to the previous structure of rTRPV2 bound to CBD (PDB:6U88) and shows an equivalent non-protein density within the CBD site. A smaller fraction of our particles represents a newly observed conformation (conformation B), which nonetheless also has a non-protein density within the previously identified CBD binding site that is fully consistent with the assigned ligand. Together, our observations provide strong support for the previously proposed location of the CBD binding site, which was one of our stated goals for obtaining structural information.

Our map of conformation B contains an additional non-protein density that we assigned to CBD. We have now expanded Figure 3 to more clearly show the two densities we assigned to CBD and show how the density is better fit by CBD than 2-APB. As pointed out by reviewer 2, the two densities we assign to CBD in conformation B are higher quality compared to conformation A (and when compared to previously published CBD densities) In the revised manuscript text we now more clearly articulate why we think the new density corresponds to CBD.

We also observed potentially interesting non-protein densities at other sites in the protein, which we did not assign, including a density within the S1-S4 domain. We agree that having our own apo structures might have allowed us to assign some of these densities. However, we carefully inspected all the previously published maps and found that contrary to the CBD-assigned density in conformation B, which is unique to our map, the rest of the unassigned densities can also be observed in several published TRPV2 structures, including an apo map (PDB:6U84, EMD-20677), a CBD-bound map (PDB:6U8A, EMD-20686) and a 2-APB bound map (PDB:7N0M, EMD-24109), suggesting that they do not represent CBD or 2-APB. We have modified Figure 3 – Figure Supp 5 to add a direct comparison with previous structures for the extra density within the S1-S4 domain, which are indistinguishable from ours.

Finally, we agree with the reviewer that with our dataset it is not possible to establish whether the distinct structural properties of conformation B were induced by binding to CBD or 2-APB. However, we argue that structures without ligands or with single ligands would also not address this conclusively: conformation B was not observed in previous structural studies of TRPV2 in the presence of 2-APB and CBD, so if we obtained structures in the apo state or in the presence of single ligands and failed to observe conformation B, this would not preclude that conformation B can be adopted under either of those conditions. Conversely, observing conformation B in the presence of just one of the ligands would not rule out that binding of the other ligand can also stabilize conformation B. Importantly, both conformations A and B are likely to represent desensitized states; regardless of the mechanistic origin of their structural differences, the general significance of conformational changes between desensitized states remains almost entirely unknown. Addressing this last point is beyond the scope of our study.

• As one of the TRPV2 mutation sites (V532M) showed no response to either 2-APB or CBD, the authors should do controls to determine via other stimuli whether the loss of response is specific to these drugs or due to a dead channel.

We agree with the reviewer that our data does not allow us to determine whether V532M channels can still respond to other stimuli. However, we consider that having that information would not have an impact on the main conclusions of the manuscript; our data establishes that position V532 is very important for channel operation, either by controlling a key step in the folding or trafficking process, or for responding to 2-APB and CBD once channels localize to the membrane. Regardless, our results establish that the identity of the residue at this position is not responsible for determining the difference in sensitization strength between rTRPV1 and rTRPV2 channels; a mutation in the equivalent position in TRPV1 results in a dramatic decrease in the apparent affinity for 2-APB, and a lesser, yet measurable reduction in the sensitivity to capsaicin, without an effect on channel trafficking. Importantly, the M572V mutation in TRPV1 does not increase the strength of sensitization by CBD, but rather further decreases sensitization.

Technical/Presentation Issues:• As a technical note for the structural work, the pixel size of the cryo-EM maps is incorrect. When compared to the 2019 and 2022 maps of TRPV2 with CBD and maps of apo TRPV1, the maps from this paper were noticeably larger. A crude scaling based on a structure of the TRPV2 ankyrin repeat domains obtained by x-ray crystallography suggests that the correct pixel size is around 0.83 Å/pix rather than 0.86 Å/pix, a significant difference in scale that needs to be corrected.

Thank you for catching this. We had used an earlier calibration and the more recent calibration of 0.83Å/pix is indeed correct.

• The authors have assigned the extra densities in the vanilloid pocket in their conformation B as a π lipid and CBD. They should provide figures showing the fit of these molecules as well as other possibilities, specifically other lipid headgroups such as PC for the lipid density and 2-APB for the CBD density. Given the conical rather than flat shape of the headgroup density and the incorrect scale of the map, it seems unlikely that the lipid in conformation B is PI.

After obtaining the new map under 0.83 Å/pix, we found that the density for the headgroup of lipid resembled a PC lipid rather than a π lipid. We have added the requested figure as Figure 3E and F. Our assignment is tentative, and we have rewritten this section to be clear that another type of lipid could be bound to this site.

Suggestions:• This paper seems to be pulling in two directions at once. On one hand, the stated goal of the paper is an investigation into the mechanisms of TRPV2 sensitization by CBD, on the other hand, the authors seem primarily interested in investigating it only from the perspective of how the group of TRPV1-TRPV3 responds to the same form of sensitization. I think the authors should pick one direction and stick with it. If they want to focus on TRPV2, they could examine TRPV2 orthologs and/or examine how CBD sensitization interacts with the other reported forms of sensitization. Alternatively, they could do a more in-depth comparative examination of the response of TRPV1-3 to CBD and 2-APB, ideally including more structural work. With either of these approaches, even if the authors are ultimately unable to track down the exact molecular basis of sensitization, they would at least have produced data that can be meaningfully interpreted for future studies on the topic.• Throughout the entire paper, the authors use a single orthologue of three different channels, rat for TRPV1 and TRPV2, and mouse for TRPV3. If the author's interest is in tracking down the mechanisms of channel sensitization, it might be a more productive approach to examine several orthologues of the same channel to see if there are varying degrees of sensitization by CBD, a feasible possibility as TRPV2 orthologs are known to respond to 2-APB differently. That would significantly reduce variables when introducing mutations to test the effect on channel sensitization. An approach like this was successfully used to examine heat desensitization of TRPV1 (Luo, Wang, Li, et al., Nat Commun 2019).• Zubcevic et al., eLife 2019 did a very nice job of tracing channel hysteresis to the conformation of the C-terminal loop and showed that the channel could be rendered completely sensitized to 2-APB by disrupting a single salt bridge with either K169A or E751A. This was tied to structural work showing a change in channel conformation, and the presence of density for 2-APB, in a structure with that mutation. This work is directly relevant to the observations in this paper, particularly as TRPV2 conserves those two residues and the conformational range at the C-terminus as observed in TRPV3. The authors may wish to investigate how this mutation interacts with 2-APB and CBD sensitization – is TRPV2 sensitized to the same extent as TRPV3 with this mutation? Does CBD still further sensitize either TRPV2 or TRPV3 with this mutation?• Work in Mo et al., eLife 2022 showed that Tyr phosphorylation can also sensitize TRPV2. The authors should mention this in their paper and may wish to investigate how this form of sensitization, as well as the methionine oxidation sensitization they already mention in the text, interacts with CBD sensitization.

These are all excellent suggestions for extending our work. We agree with some of the stated shortcomings of our study, and we have tried to revise the manuscript to improve/clarify the presentation and we have commented on this reviewer’s idea that differences in the binding site for 2-APB could be responsible for the different sensitization strength observed between TRPV1 and TRPV2 channels. With all due respect however, we think our study provides valuable new findings that need to be considered in future studies. Our structural data alone are the highest quality data for CBD, we identify a real issue concerning previous assignments of 2-APB, by pointing towards densities that are also present in apo datasets in the absence of 2-APB, and we provide a thorough characterization for how CBD sensitizes TRPV channels. The sensitizing effect of CBD on TRPV1 and TRPV3 channels had not been previously reported. In this regard, we want to bring attention to the robustness of our observations and the magnitude of the effect that we describe: the sensitization caused by CBD in rTRPV2 and mTRPV3 is massive, whereas that observed in TRPV1 is at least one order of magnitude weaker. Robust effects like this are ideal for mechanistic investigation because interpretation of results can be unambiguous – we therefore think that focusing on the differences that we highlight here between the three channels offers a promising path for uncovering fundamental differences in the way each of these channels work. Our data already establishes that the CBD site is not responsible for the differences in sensitization strength between the distinct TRPV channels, and that other sites, possibly including the 2-APB interaction site(s), must be responsible for the observed differences. In addition, our observations are relevant because of the limited number of available stimuli that can robustly activate TRPV2 and TRPV3 channels, which otherwise require 2-APB concentrations close to its solubility limit to reach maximal activation.

Finally, we also recognize that rTRPV2 channels have multiple mechanisms of sensitization (heat, oxidation of methionine residues, phosphorylation, likely others) in addition to the one we describe here caused by CBD. The allosteric nature of these receptors strongly suggests that each of these sensitizing mechanisms will have various degrees of mechanistic overlap (in the end, each one of them will affect opening and closing of the pore gate) – disentangling these mechanisms in terms of defined conformational changes is one of the current goals of our research program. However, we are aware of the complexities that this would entail, and that the amount of work required to achieve this goal is well beyond the scope of any singular publication.